# INSTRUCTDET: DIVERSIFYING REFERRING OBJECT DETECTION WITH GENERALIZED INSTRUCTIONS

**Ronghao Dang**[1*]   **Jiangyan Feng**[2*]   **Haodong Zhang**[2]   **Chongjian GE**[3]   **Lin Song**[4]
**Lijun Gong**[2]   **Chengju Liu**[1]   **Qijun Chen**[1]   **Feng Zhu**[2]   **Rui Zhao**[2]   **Yibing Song**[5★]
[1]Tongji University   [2]SenseTime Research   [3]The University of Hong Kong
[4]Tencent AI Lab   [5]Alibaba DAMO Academy

dangronghao@tongji.edu.cn   fengjiangyan@sensetime.com   yibingsong.cv@gmail.com

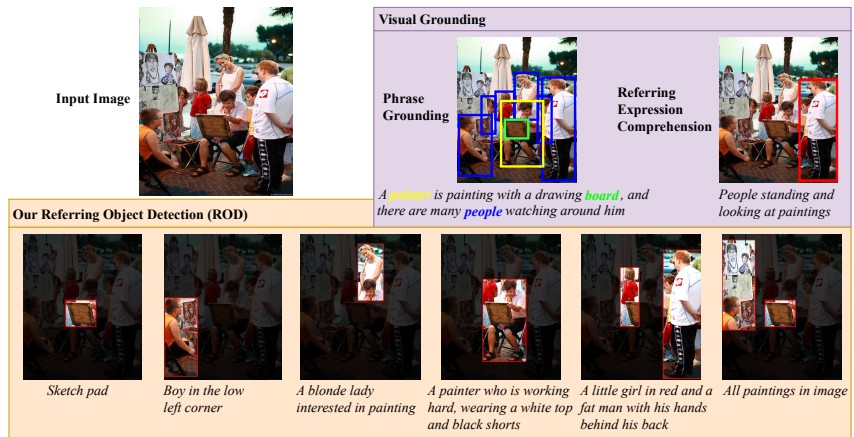

Figure 1: Our ROD aims to execute diversified user detection instructions compared to visual grounding. For images with object bbxs, we use foundation models to produce human-like object detection instructions. By training a conventional ROD model with incorporating tremendous instructions, we largely push ROD towards practical usage from a data-centric perspective.

## ABSTRACT

We propose InstructDET, a data-centric method for referring object detection (ROD) that localizes target objects based on user instructions. While deriving from referring expressions (REC), the instructions we leverage are greatly diversified to encompass common user intentions related to object detection. For one image, we produce tremendous instructions that refer to every single object and different combinations of multiple objects. Each instruction and its corresponding object bounding boxes (bbxs) constitute one training data pair. In order to encompass common detection expressions, we involve emerging vision-language model (VLM) and large language model (LLM) to generate instructions guided by text prompts and object bbxs, as the generalizations of foundation models are effective to produce human-like expressions (e.g., describing object property, category, and relationship). We name our constructed dataset as InDET. It contains images, bbxs and generalized instructions that are from foundation models. Our InDET is developed from existing REC datasets and object detection datasets, with the expanding potential that any image with object bbxs can be incorporated through our InstructDET method. By using our InDET dataset, we show that a conventional ROD model surpasses existing methods on both standard REC datasets and our InDET test set. InstructDET, our data-centric method with automatic data expansion by leveraging foundation models, directs a promising field that ROD can be greatly diversified to execute common object detection instructions.

---

*R. Dang and J. Feng contribute equally. ★Y. Song is the corresponding author. This work is done when R. Dang is an intern at Sensetime. The code is available at https://github.com/jyFengGoGo/InstructDet.

# 1 INTRODUCTION

Referring object detection (ROD) aims to detect target objects according to language reference that represents user intentions. ROD is closely related to visual grounding where there are phrase grounding (Akbari et al., 2019; Li et al., 2022a; Gao et al., 2023) and referring expression comprehension (Su et al., 2020; Zhu et al., 2022). As shown in Fig. 1, phrase grounding detects all objects mentioned in one sentence, while referring expression comprehension (REC) only detects one single object that the text refers to. As such, the language reference in REC shall be discriminative and specifically relates to one object without ambiguity.

Currently, visual grounding develops at an initial stage and leaves a gap for practical usage. The phrase grounding does not differentiate which object ought to be detected via language description, while REC only targets for one object with single text reference. In the current REC datasets, each image contains few expressions (e.g., 1 or 2 phrases). These expressions are insufficient to represent user intentions. In an image where there are several objects, users may want to detect each single object by using different descriptions (e.g., object color, shape, or location), or detect multiple objects in different combinations (e.g., similar properties or relationships). These diverse expressions are not conveyed within current REC datasets, leaving the gap for existing methods to practically fulfill user intentions for visual grounding. Moreover, the manual collection of these expressions are cumbersome, and subject bias prevents an effective coverage of common user intentions when perceiving each image. Therefore, the practical user expressions are not well fulfilled when they expect to detect various objects in one image.

In this work[1], we aim to push visual grounding toward practical usage from a data-centric perspective. Instead of developing REC models to generalize based on current data, we set up referring object detection (ROD) scenario to automatically diversify user expressions. Inspired by the generalization of foundation models that execute common user instructions based on the image and text inputs, our InstructDET borrows their capabilities to produce human-like instructions that encompass user intentions related to object detection. The generalized instructions produced by the foundation models can be regarded as an expansion of existing user expressions in REC. We produce instructions that describe single object from two pipelines. In the first pipeline (i.e., global prompt), we convert an image into an elaborate text description via LLaVA (Liu et al., 2023a). The text description, together with object bbxs coordinates, are sent to the LLaMA (Touvron et al., 2023) for instruction generation in global prompt. During generation, we manually write 3 in-context examples and leverage the in-context learning (Dong et al., 2023) ability of LLaMA to describe the content related to each object following the format of our examples.

In the second pipeline (i.e., local prompt), we send the image and text prompts into LLaVA. The objects in the image are marked with bbxs and the text prompts require LLaVA to describe the object content. We initialize LLaVA with miniGPT4 weights and find it tends to produce lengthy and global descriptions. So we perform a partial finetuning on LLaVA by using REC data to let it focus on local objects. Through these two pipelines, we observe that instructions generated from global prompt pipeline focus more on the object relationship, while instructions generated from local prompt pipeline focus more on rich visual details and advanced logic reasoning. Naturally, we combine instructions from these two pipelines to formulate expressions for single referred object. During instruction generation, the uncontrolled model hallucination (Li et al., 2022b) brings incorrect or irrelevant instructions. We propose to use visual-textual verification via CLIP (Radford et al., 2021) for effective instruction filtering.

The generalization and reasoning of foundation models (Wang et al., 2022; Zhou et al., 2022) provide sufficient instructions encompassing user intentions for single object description. When describing multiple objects, we divide descriptions into two parts. The first part is to independently describe each single object followed by concatenation, and the second part is to summarize commonalities of multiple objects. The commonality summarization requires unifying similar or related objectives by a higher-level language abstraction that describes their similarities and relationships. We collect the

---

[1]We do not differentiate "instruction" and "expression" in this paper, as both of them represent user intentions. For presentation clarity, in our InstructDET pipeline we refer expressions that are generated by foundation models, and we further refine expressions to instructions for InDET inclusion. As we only focus on ROD, we can formalize our instruction by simply adding the word 'detect' beforehand.

combinations of different objects via semantic clustering, then utilize LLM to generate commonality summarizations for each combination.

We automatically collect instructions targeting for single or multiple objects in images and construct our InDET dataset. Sec. 4 shows an in-depth analysis of our dataset where we establish a guideline to organize these instructions from 6 aspects. Compared to existing REC datasets where the instructions only reside in sub-parts of our groups, our InDET is more comprehensive to incorporate user intentions of object detection. Fig. 1 shows an intuitive example of the generalized expressions produced by foundation models. By using our InDET dataset, we train a conventional ROD model and find it surpasses existing VG models on standard benchmarks and our InDET test set. Moreover, we also validate that our model has learned to effectively understand the meaning of instructions rather than only recognize key words, which is because of the tremendously expressive instructions incorporated for our model training. Our InstructDET method can automatically expand training data by using in-the-wild images with object bbxs, which improves our model generalizations towards practical usage. In addition, our model can already serve as the detection module of the neural-symbolic visual compositional task solution given arbitrary language instructions beyond object detection (e.g., Visual ChatGPT (Wu et al., 2023), VISPROG (Gupta & Kembhavi, 2023)).

## 2 RELATED WORKS

**Visual Grounding**. Studies on visual grounding (Kamath et al., 2021; Chen et al., 2021; Deng et al., 2021; Su et al., 2023) can be mainly categorized as phrase grounding (Plummer et al., 2022; Kojima et al., 2023) and REC (Hudson & Manning, 2018; Li & Sigal, 2021). Phrase grounding detects all objects mentioned in the text while REC localizes one object that the text referred to. In (Zhang et al., 2022; Liu et al., 2023c), the objects mentioned in the text are verified to each visual object proposal one-by-one. These methods require a clear and specific object referring in the text. On the other hand, methods (Zhu et al., 2022; Yan et al., 2023) based on DETR (Carion et al., 2020) can accept abstract and summarized descriptions such as "red objects" and "all objects". Our ROD model follows DETR-based design to enrich interpretation of various instructions. Note that our model is learned via InDET dataset where instructions are produced based on preset object bbxs.

**Referring Expression Datasets**. The REC datasets are usually constructed via manual annotation on the images. A two-player game is utilized in (Kazemzadeh et al., 2014) where the text descriptions are concise due to limited relevant visual contents. The RefCOCO, RefCOCO+ (Yu et al., 2016), and RefCOCOg (Mao et al., 2016) employ MSCOCO (Lin et al., 2014) images for manual expression production. The expression flexibility and diversity of these datasets are limited to encompass common detection intentions. Recent datasets (Krishna et al., 2017; Kuznetsova et al., 2020; Kebe et al., 2021) focuses on data scalability rather than auto generation. Cops-Ref (Chen et al., 2020) leverages scene graph as reasoning groundwork, thus forming a tree structure to generate expressions with varying compositionality. Different from these methods based on template guided expression generation, our InstructDET relies on foundation models to produce well generalized and human-like instructions.

**Data Generation via Foundation Models**. The InstructGPT (Ouyang et al., 2022) and GPT4 (OpenAI, 2023) have shown generalization and reasoning abilities for data generation. LLaVA (Liu et al., 2023a) first uses GPT4 for multi-modal data generation following instructions. Otter (Li et al., 2023a) performs multi-modality in-context instruction tuning by levering multiple images, questions, and answers. Currently, these models focus on global image and language understanding, with less focus on local object analysis. Moreover, these multi-modality models, although processing multi-modality data, still outputs single-modality text description. There is a gap for these foundation models to function in the computer vision scenarios, especially visual recognition. In comparison, our InstructDET uses foundation models to benefit ROD model training, which contributes directly to improve object detection performance.

## 3 INSTRUCTDET

Fig. 2 shows an overview of our InstructDET method for data construction. Given an input image with object bbxs, we use two pipelines to produce detection expressions from foundation models. The expressions are further refined to instructions and incorporated into our InDET dataset.

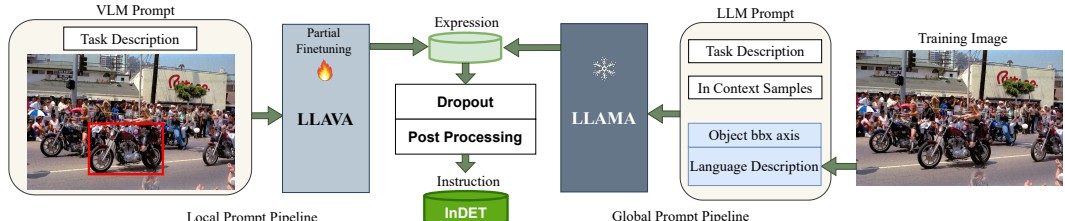

Figure 2: An overview of our InstructDET. We use two pipelines to produce detection expressions via foundation models. In the global prompt pipeline, we use LLaVA to describe an image via text, and combine this text with other text prompts for LLaMA input. In the local prompt pipeline, we use the same image with object bbxs and text prompts as multi modality input for LLaVA. The produced expressions are further refined to instructions and incorporated into our InDET dataset.

## 3.1 GLOBAL PROMPT PIPELINE

The large language model (LLM) has shown surprising generalizations to well execute common user instructions. We use LLM to simulate user intentions when perceiving objects in an image. Our global prompt pipeline produces a text prompt for the LLaMA[2] model. This prompt consists of several contents including global image description, object bbx coordinates, in-context samples, and task description. Without tuning LLaMA, we obtain instructions that describe objects in an image. A detailed example is shown in Sec. C for an intuitive illustration of how we leverage foundation models to produce expressions. We elucidate the key steps during this process as follows:

Given an image with object bbxs, we first obtain global image description in text form. If this image already contains dense captions (e.g., from Flicker30K), we directly load these captions. Alternatively, we leverage LLaVA to generate the global image description. The text prompt we use for LLaVA contains our language guidance to emphasize that specific interested object categories shall be mentioned. As such, LLaVA will describe each labeled object in its output. As for object bbx content, if the image is from REC dataset, we use referring expression as the object content. Otherwise, we simply use the category name.

When designing the task description prompt, we expect LLaMA to produce diverse expressions that contain different properties of single object as much as possible. We manually list the attributes from the user perspective, including the object type, color, function, motions, etc. Besides, we include the object attributes of its relationship with other objects in image, such as object interactions, object relative positions, etc. When using these aforementioned prompts for LLaMA input, we find that the output text varies significantly and might be irrelevant to the target objects. Inspired by the in-context learning ability of foundation models, we manually design in-context samples to regularize the output content and format. The output results will thus resemble our in-context examples but with our expected diversified object descriptions.

## 3.2 LOCAL PROMPT PIPELINE

The global prompt pipeline produces expressions according to text prompts. Naturally, we can feed both image and text prompt to the multi-modality foundation model for object description. Given an image, we mark the object with bbx rectangle, and send this image to LLaVA, together with the text prompt that requires LLaVA to describe the object according to the bbx. Here, the bbx serves as a visual highlight for LLaVA to comprehend the target object that we expect to describe.

Our LLaVA model is initialized with miniGPT4 weights. When we send these multi-modality inputs to LLaVA, we observe that LLaVA produces detailed and dense descriptions for the image, rather than expressions of the specific target object. We analyze that the vision-language alignment module in LLaVA is the Q-Former (Li et al., 2023b), which transforms one image into only 32 visual tokens without concentrating on the local objects. Meanwhile, LLaVA itself tends to produce lengthy and dense descriptions. In order to generate instructions suitable for ROD, we finetune a part of LLaVA by using existing REC datasets. Specifically, we only update a linear layer that transforms visual

---

[2] In this paper, we use a variant of LLaMA (i.e., Vicuna 13B) that has gone through instruction tuning. Besides, we use a variant of LLaVA, which is a multi-modal paradigm that maps visual features into token embeddings with further alignment to the text domain.

tokens to the text embedding space during training. The linear layer is learned to attend local objects with concise expressions. After finetuning, we observe the LLaVA output becomes informative and closely related to the target object. Detailed examples of generating expressions in local prompt pipeline are shown in Sec. B, and the detailed analysis on how finetuning improves LLaVA output is provided in Sec. H.

### 3.3 EXPRESSION FILTER

In global and local pipelines, we have regularized the output of foundation model from several aspects including text prompt specification, in-context learning, and model finetuning. In practice, we still observe the model hallucination phenomena that the model sometimes generate expressions describing objects not even exist in the image. Moreover, the expressions from the local prompt pipeline sometimes describe the whole image rather than local objects. This is due to the miniGPT4 initialization of LLaVA, which utilizes dense captions for instruction tuning. The tendency to generate global image description is mitigated via our model finetuning to focus on local object, but not completely disappeared. To further improve the expression quality, we introduce visual and language matching via CLIP (Radford et al., 2021) to filter out inappropriate expressions. Fig. 3 shows an overview. It contains image visual prompting and visual-textual matching.

**Visual Prompting** We study visual language pretraining (VLP) (Yang et al., 2023; Shtedritski et al., 2023) where visual prompting is developed for images. We observe that in zero-shot REC, coupling VLP with visual prompts enables robust pairing of local image region and corresponding text description. In the pairing process, the design of visual prompting heavily influences the visual-textual matching results. Specifically, we employ the superposition of a red ellipse and the target Gaussian blur reversion as visual prompts. A detailed pipeline illustrating visual prompting is in Sec. D.

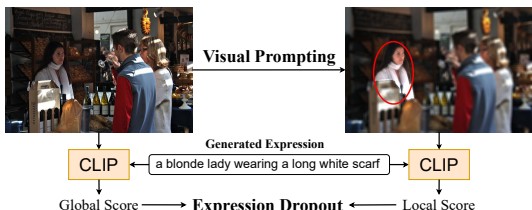

Figure 3: Expression filtering by image visual prompting and visual-textual matching via CLIP.

**Visual-Textual Matching**. We use images with visual prompting that emphasizes target objects to verify the corresponding text descriptions via a frozen CLIP model. While local object contents are well aligned with target referring expressions, we observe that expressions describing the whole image are not eliminated by CLIP. We analyze that CLIP is originally trained to focus on the correlations between global image features and global textual semantics. The global visual-textual matching makes CLIP model to prefer global image description accordingly. To remove this effect, we establish a referring measurement from both local and global perspectives. For the image in Fig. 3, we compute a global score $S_g$ and a local prompt score $S_l$. The magnitude of referring expression can be measured via our local enhancement score $S_e = S_l - S_g$. Our final expression evaluation score can be computed as:

$$S_f = \alpha_1 S_e + \alpha_2 S_l = (\alpha_1 + \alpha_2)S_l - \alpha_1 S_g = S_l - \alpha_1 S_g \tag{1}$$

where $\alpha_1$ and $\alpha_2$ are scalars balancing the contributions of $S_g$ and $S_l$ with $\alpha_1 + \alpha_2 = 1$. So $\alpha_1 \in [0, 1]$ adjusts the final score towards local content referring or global semantics. Note that we introduce $S_e$ to measure difference between local and global scores. If the expression is more related to the target object, $S_e$ becomes higher after visual prompting for object highlights. After computing $S_f$, we set a dynamic threshold to filter out expressions. This is because $S_f$ is based on CLIP's preference that a small target object with well matched expression achieves a lower score than a large object with mismatched expression. Therefore, we use provided expression (for images from REC) or category name (for images out of REC) to compute a final score, and discard generated instructions whose $S_f$ is lower than this score.

### 3.4 MULTI-OBJECTS EXPRESSION GENERATION

Our expression generation pipeline illustrated above targets for each object independently. In practice, users may refer to multiple objects in one image. We study common user expressions for multi-objects, and conclude them into two aspects. The first one contains splicing expressions that

combine different single object expressions with 'and' or comma. In this case, the objects mentioned in the expression are not related to each other. The second aspect contains generalization expressions that summarize the common properties of multiple objects (e.g., color, category, or location) to produce an abstract and conceptual description. It resembles mining similarities between multiple objects and thus is not straightforward to conclude. Therefore, we need to discover object combinations where similar properties may exist, and then summarize the commonalities among them to constitute the summarized expressions.

Our process to produce summarized expression is shown in Fig. 4. For each object, we first concatenate all its related expressions with commas. Through this concatenation, we can obtain this object expression from different perspectives (i.e., different properties). Then, we use the text encoder in BERT (Devlin et al., 2018) to map this concatenated expression to a semantic embedding space. As such, we obtain embeddings of concatenated expressions from all objects. Then, we cluster these embeddings into indeterminate number of clusters by using DBSCAN (Ester et al., 1996) method. We use

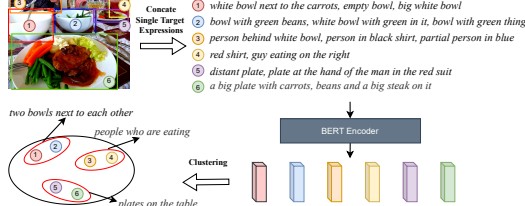

Figure 4: Mining commonalities among multi-objects via expression concatenation and text semantic clustering, followed by LLaMA descriptions on each cluster center.

LLaMA to generate text for clusters with multiple objects. The details of using LLaMA to mine object commonalities are in Sec. E. The generated text indicates the summarized expression we aim to produce for multiple objects.

**Post Processing**. After generating expressions for single and multiple objects, we verify and remove the repeated expressions that pertain to the same object. Then, we utilize LLaMA to further diversify these generated expressions while preserving their original intents, i.e., we use LLaMA to do synonymous rewriting of generated expressions. The prompt we use for synonymous rewriting is provided in Sec. I. We observe that for different objects in one image, the expression for one object may be similar to that of others. These expressions are ambiguous since we can not refer to a unique object based on their referring. Nevertheless, we transfer these expressions to refer to multi-objects since they express a set of objects in one image. This transfer further augments multi-object referring expressions. Finally, we collect these remaining expressions after post processing as instructions. Together with corresponding object bbxs and images, we construct our InDET dataset by incorporating diversified object detection instructions encompassing user intentions.

## 4 DATASET ANALYSIS

Our InDET dataset contains images from MSCOCO (Lin et al., 2014), Flicker (Plummer et al., 2015), and Objects365 (Shao et al., 2019). There are 120.6K images with 908.4K referring object sets in total. Together with original expressions, there are 3.6M instructions in total, making InDET the largest real-world REC dataset at present. The average instruction length is 6.2 words and the vocabulary size is 63k words, which surpasses existing automatically annotated datasets in terms of instruction quantity, richness, and vocabulary breadth. We split the images into training, validation, and testing sets, with the corresponding instruction amount of 3139K, 240K, and 247K, respectively. In the following, we first propose a guideline that represent common user intentions and divides existing instructions into 6 groups. Then, we analyze all the instructions in InDET according to this guideline to show how our InDET advances REC scenario compared to existing datasets.

**Instruction Guideline**. The instructions in InDET dataset describe objects from various perspectives. We observe that these descriptions all focus on object category, attribute, and relations, but with different emphasis extent. Based on expression complexity, we establish a guideline that divides all instructions into 6 groups. Each group reflects one level of emphasis on category, attribute, and relations. Table 1 shows our guideline and examples. The first four groups are for single object and the last two groups are for multiple objects. In the first group (G1), there is only one single phrase to describe object category, which is similar to the traditional object detection task. From G2 to G4, more phrases are involved to describe the target object. For G5, we construct a spliced form to combine instructions from different single objects. In G6, the instruction is a general description

Table 1: Instruction guideline and samples. Our guideline contains 6 aspects and covers common user intentions. These aspects are built upon object category, attribute, and relations with different emphasis levels. We use ⋆ and ⋆⋆ to indicate the description complexity of different aspects.

| Aspect | | Category | Attribute | Relation | Examples |
|---|---|---|---|---|---|
| Single Object | 1 | ⋆ | | | *pencil; two children; soccer ball; city street* |
| | 2 | | ⋆ | | *shirts with English letters; red and white airplane* |
| | 3 | | ⋆⋆ | ⋆ | *man in blue shirt halfway on screen; people who are sitting under an umbrella; a man in a grey sweater and black jeans performing a skateboarding trick;* |
| | | | ⋆ | ⋆⋆ | |
| | 4 | | ⋆⋆ | ⋆ | *a woman sitting cross-legged on the couch with her back facing the viewer. She has a white shirt and black pant; the glasses are sitting on top of some kind of paper or folder and there is a book and a lantern next to it* |
| | | | ⋆ | ⋆⋆ | |
| | | | ⋆⋆ | ⋆⋆ | |
| Multiple Objects | 5 | Single object combination | | | *a black hat on a man's head and red umbrella and blue truck in rains* |
| | 6 | Commonality generalization | | | *every object on table; kids playing with the blond boy* |

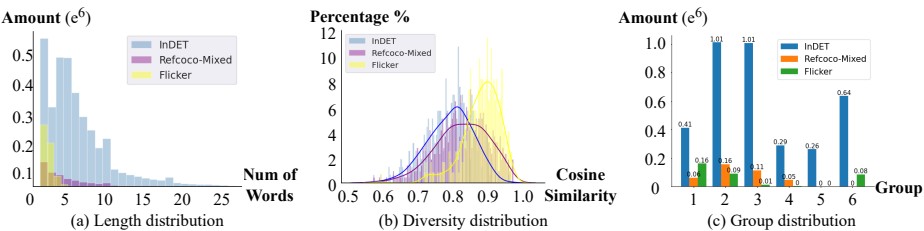

Figure 5: Dataset analysis of expression length, diversity and group distributions.

(a) Length distribution  (b) Diversity distribution  (c) Group distribution

of commonality between multiple objects. To this end, the instructions from G1 to G6 gradually introduces semantic understanding, visual language grounding, and logic reasoning for ROD. After guideline construction, we use LLaMA to assign each instruction into our groups by using in-context learning that let LLaMA to understand assigning principles and in-context assigning examples. The detailed usage of LLaMA for instruction assign is shown in Sec. F.

**Instruction Length, Diversity, and Aspect Ratio Distributions**. We analyze our InDET from the instruction length, diversity, and ratio distribution in our guideline groups. The RefCOCO and Flicker datasets are introduced for comparison. Fig. 5(a) shows the number of word distribution where the instruction of InDET contains more words than the other two datasets. Moreover, there are 100K instructions in our InDET consist of more than 10 words, while other datasets do not contain such informative expressions. In Fig. 5(b), we show diversity comparison where we use CLIP to map all instructions into a semantic space. Then, for the same target objects we compute average pairwise cosine similarity. The results show that our InDET contains lower value than other datasets, which indicates that our instructions are more diverse when describing the same target object. In Fig. 5(c), we show aspect ratio distribution of expressions assigned by our guideline. For existing datasets, user expressions commonly exist in G1 and G2. In contrast to the user expressions that seldom exist from G3 to G5 for Flicker, and seldom exist in G5 and G6 for RefCOCO, the instructions in our InDET exist normally in all groups. This distribution shows that our InDET is more effective to encompass common user intentions, especially for multiple objects. By leveraging our InDET, the ROD model becomes more practically applicable.

## 5 REFERRING OBJECT DETECTION

In this section, we illustrate our model design for ROD task. We notice that ROD shares little difference with visual grounding (VG). First, ROD produces uncertain number of object bbxs (i.e., 0, 1, or multiple) based on one input instruction, as shown in Fig. 1. Second, ROD supports abstract and summarized object descriptions (e.g., "all objects on the table") that do not clearly refer to specific objects such as "bottle", "orange", and "knife". As recent VG models (Zhang et al., 2022; Liu et al., 2023c) require a one-by-one verification between visual objects and expression words, they are not able to execute such instructions. Motivated by the difference, we set up a conventional framework from DETR-based VG models (Zhu et al., 2022; Yan et al., 2023). Fig. 6 shows an overview of our DROD model. We illustrate key steps as follows:

Given an image with text instruction, we use visual and text encoders (Dosovitskiy et al., 2021; Devlin et al., 2018; Ge et al., 2023) to obtain their embeddings. Then, we use a bi-directional cross

attention module to perform multi-modality embedding fusion. For the fused visual embedding, we sent it to the transformer encoder and decoder structure (Zhu et al., 2020) with $N$ learnable queries as position priors (Meinhardt et al., 2022; Ge et al., 2021). Then, the decoder produces $N$ instance proposals for further selection. For the fused text embedding, we pass it through a global average pooling and MLP for text2visual embedding space mapping. Finally, we use cosine similarity to match proposals and mapped text embedding. During the training stage, we use confidence loss and localization loss via supervised learning. During the inference stage, we select proposals whose matching scores are above a predefined threshold, which allows our model to produce arbitrary number of bbxs for diversified instruction execution. More details are shown in Sec. G.

## 6 EXPERIMENTS

We evaluate the ROD performance on standard VG benchmarks (i.e., RefCOCO, RefCOCO+, and RefCOCOg) and our InDET dataset. As illustrated in Sec. 4, the images with marked objects of our InDET dataset are collected from existing datasets while the instructions are significantly enriched. We split the training and test set of InDET following RefCOCO/g/+ where the test set contains 6.5k images with an increased number of instructions to 315K. Moreover, these instructions are assigned to 6 groups according to our guideline. The perfor-

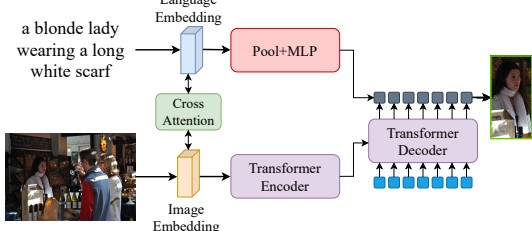

Figure 6: An overview of our diversified referring object detection (DROD) model.

mance on each group reflects how VG methods perform when processing different aspects of user instructions. The comparing methods in our experiments are from recent VG methods including MDETR (Kamath et al., 2021), Grounding-DINO (Liu et al., 2023c) and UNINEXT (Yan et al., 2023). Due to page limit, we show the evaluation results on our InDET, our InDET with shuffled expression, and standard benchmarks. Model training, and ablation studies on partial LLaVA finetuning and visual prompt selection are provided in Sec. G and Sec. H. We also provide visual comparison results of these methods in Sec. A. A video demo showing the practical usage of our DROD model is in our webpage.

Table 2: Evaluation results on our InDET and shuffled InDET test sets. We show the object bbx average precision (AP) values (%) of these two test sets with a slash ('/') separation.

| Method | Backbone | AP | AP by Group | | | | | |
|---|---|---|---|---|---|---|---|---|
| | | | G1 | G2 | G3 | G4 | G5 | G6 |
| MDETR | ResNet101 | 34.86 / 31.21 | 47.44 / 46.61 | 46.79 / 42.55 | 34.14 / 28.13 | 23.22 / 16.86 | 25.91 / 23.52 | 28.17 / 23.66 |
| G-DINO | SwinB | 35.96 / 30.43 | 47.10 / 45.91 | 47.17 / 42.56 | 35.29 / 27.28 | 26.84 / 18.46 | 27.95 / 23.74 | 27.61 / 23.57 |
| UNINEXT | ResNet50 | 43.37 / 37.61 | 54.49 / 53.09 | 54.52 / 49.91 | 44.49 / 35.59 | 37.17 / 28.30 | 31.41 / 28.28 | 32.01 / 27.52 |
| DROD (Ours) | ResNet50 | 62.24 / 53.78 | 67.14 / 65.08 | 67.34 / 61.56 | 60.89 / 48.82 | 55.10 / 41.50 | 70.15 / 64.64 | 74.22 / 67.11 |
| DROD (Ours) | ViT-H | 66.90 / 57.32 | 72.53 / 69.79 | 72.47 / 65.44 | 66.42 / 52.50 | 59.86 / 46.01 | 73.34 / 67.82 | 75.92 / 68.73 |

In our InDET test set, we compare our DROD model to other methods under the evaluation metric of object bbx average precision with a threshold of 0.5. On the other hand, we investigate whether these methods have truly comprehended the meaning of instruction, or they perform ROD only based on the key words (e.g., noun) without comprehending the whole expression. So we shuffle the InDET test set by randomly ordering the words in each instruction. We produce results of existing VG methods on our InDET test set without assuming object numbers in advance. For one method, if its performance drops more on the shuffled data, this method is shown to better comprehend the meaning of instruction.

Table 2 shows the evaluation results. Overall, UNINEXT achieves a higher AP than MDETR (i.e., 43.37 v.s. 34.86) in our InDET test set, while decreasing more than MDETR (i.e., 37.61 v.s. 31.21) in shuffled data. This indicates that UNINEXT is more effective than MDETR for ROD and better comprehends instruction meaning. Meanwhile, UNINEXT achieves a higher AP value than Grounding-DINO. In comparison, our DROD largely surpasses UNINEXT (62.24 v.s. 43.37) on the overall AP comparison, and using a VIT encoder further increases our performance. This indicates that our DROD is more effective to comprehend generalized instructions for ROD. Meanwhile, we observe that our performance drop is larger than UNINEXT (8.46 v.s. 5.76), which shows that our

Table 3: Evaluation results on the RefCOCO/g/+ datasets. We follow evaluation protocols to report AP values (%) of comparing methods. We use the notations "CC", "VG", "OI", "O365", "RIGame", for COCO, Visual Genome, OpenImage, Objects365, ReferItGame, respectively.

| Method | Backbone | Data | RefCOCO | | | RefCOCO+ | | | RefCOCOg | |
|---|---|---|---|---|---|---|---|---|---|---|
| | | | val | testA | testB | val | testA | testB | val-u | test-u |
| RefTR | ResNet101 | VG | 85.65 | 88.73 | 81.16 | 77.55 | 82.26 | 68.99 | 79.25 | 80.01 |
| SeqTR | DarkNet53 | VG,RIGame,Flickr,RefC | 87.00 | 90.15 | 83.59 | 78.69 | 84.51 | 71.87 | 82.69 | 83.37 |
| MDETR | ResNet101 | GoldG,CC,RefC | 86.75 | 89.58 | 81.41 | 79.52 | 84.09 | 70.62 | 81.64 | 80.89 |
| G-DINO | SwinB | O365,CC,RefC,GoldG,etc | 83.95 | 87.79 | 79.16 | 72.91 | 80.91 | 62.96 | 76.98 | 76.76 |
| UNINEXT | ResNet50 | O365,CC,RefC | 87.64 | 90.35 | 83.49 | 78.14 | 83.22 | 68.71 | 80.96 | 81.86 |
| DROD (Ours) | ResNet50 | O365,CC,InDET | 88.92 | 90.86 | 85.57 | 78.27 | 83.39 | 71.04 | 83.01 | 82.91 |

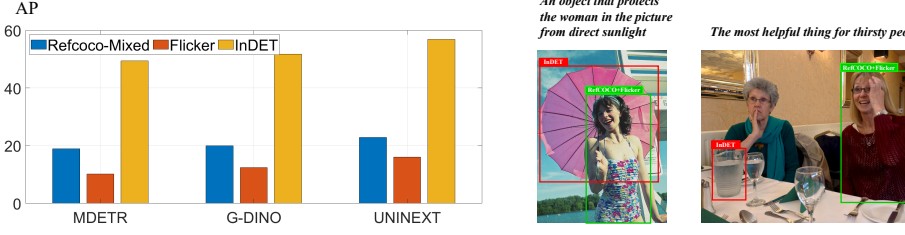

(a) Numerical Results    (b) Visual Comparisons

Figure 7: Our InDET dataset improves logic reasoning of ROD models. In (a), existing models trained with our InDET dataset show superior results compared to other datasets. In (b), we show visual comparisons by using the same DROD model but with different training datasets.

model better comprehends different expressions. Specifically for the results in each group, we notice that our performance drop is little in G1, and becomes larger from G2 to G4. This is because more and more words are introduced from G1 to G4 for object description. A random order gradually affects our model comprehension. For G5 and G6, we note that our method largely outperform other methods. The multi-object instructions incorporated in the dataset improves our performance.

Besides evaluating our InDET test set, we compare our DROD model with existing VG methods (Zhu et al., 2022; Li & Sigal, 2021) on the standard VG benchmarks RefCOCO/g/+. Table 3 shows the evaluation results. Overall, our DROD model achieves favorable performance on these datasets. This is because our DROD model utilizes InDET dataset where diversified instructions improve model generalizations. By using a conventional ROD model, we improve the VG performance from the data diversity perspective.

In addition to the overall precision comparisons, we evaluate how our dataset improves logic reasoning and instruction comprehension of existing models. Specifically, we select 2k test samples from our InDET test dataset where logic reasoning on instructions is required for object detection. For each model (i.e., MDETR, G-DINO, or UNINEXT), we train it by using different datasets (i.e., RefCOCO, Flicker, or InDET) and show the performance comparison on our 2k test samples. Fig. 7(a) shows the evaluation results where each model trained with our InDET dataset outperforms the same model trained with other datasets. In Fig. 7(b), we show visual comparisons by using our DROD model but with different training sets. It shows that using original datasets, the model tends to ground keywords rather than preform multi-modal reasoning based on instructions. In comparison, by training with our InDET, the model well interprets instruction meaning and conduct logic reasoning across languages and visual images.

## 7 CONCLUDING REMARKS

We aim to push ROD into practical usage from a data-centric perspective. On one hand, we notice that current REC expressions are insufficient to encompass user detection intentions. On the other hand, foundation models have shown promising generalizations to simulate manual understanding and description abilities. To this end, we develop InstructDET that leverages foundation models to produce human-like expressions in REC, which tends to incorporate common user intentions into ROD training. As a result, our DROD model achieves favorable performance compared to existing VG methods. In the future, we can combine our method with open-set object detectors to fully explore in-the-wild images (e.g., Internet images) for comprehensive user expression generation. We expect our DROD model to generalize as much as existing foundation models, and thus take a huge step towards completely solving ROD task.

## ACKNOWLEDGEMENT

This paper is supported by the National Natural Science Foundation of China under Grants (62073245, 62173248,62233013). Shanghai Science and Technology Innovation Action Plan (22511104900). Shanghai Municipal Science and Technology Major Project (2021SHZDZX0100) and the Fundamental Research Funds for the Central Universities.

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

APPENDIX OVERVIEW

We provide an overview to present a clear understanding of this section.

- In Sec. A, we present visual comparisons of recent VG methods under various instructions.
- In Sec. B and Sec. C, we describe the process of generating expressions from foundation models in both global prompt and local prompt pipelines, respectively.
- In Sec. D, we detail the aspects of visual prompting and visual-textual matching.
- In Sec. E, we explain our approach to designing prompts that enable LLaVA to extract commonalities among multiple objects for summarized expression generation.
- In Sec. F, we discuss our strategy for designing prompts that allow LLaMA to assign generated instructions to our predefined groups.
- In Sec. G, we provide an overview of our model architecture and implementation specifics.
- In Sec. H, we present ablation studies investigating LLaVA fine-tuning in local prompt pipeline and visual prompting selection.
- In Sec. I, we provide the prompt we used in post processing.
- In Sec. J, we add supplementary evaluation results to Table. 2 and Table. 3.
- In Sec. K, we add the time consumption of main steps in our instruction generation procedure.

## A  VISUAL COMPARISON RESULTS

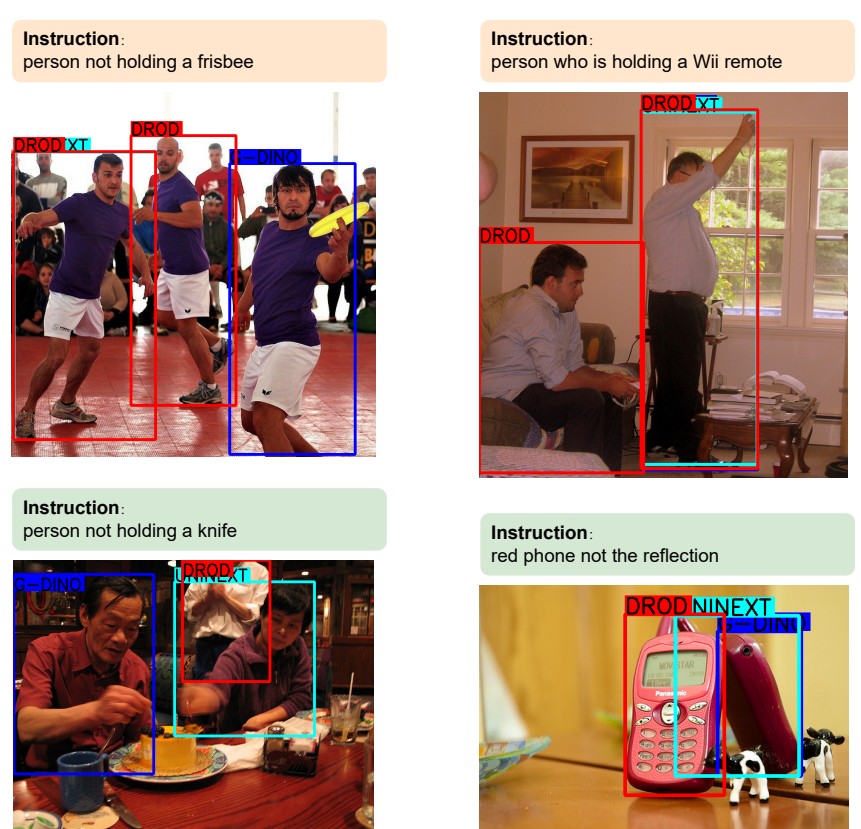

Figure 8: Visual comparison results. Our DROD well executes detection instruction compared to Grounding-DINO and UNINEXT.

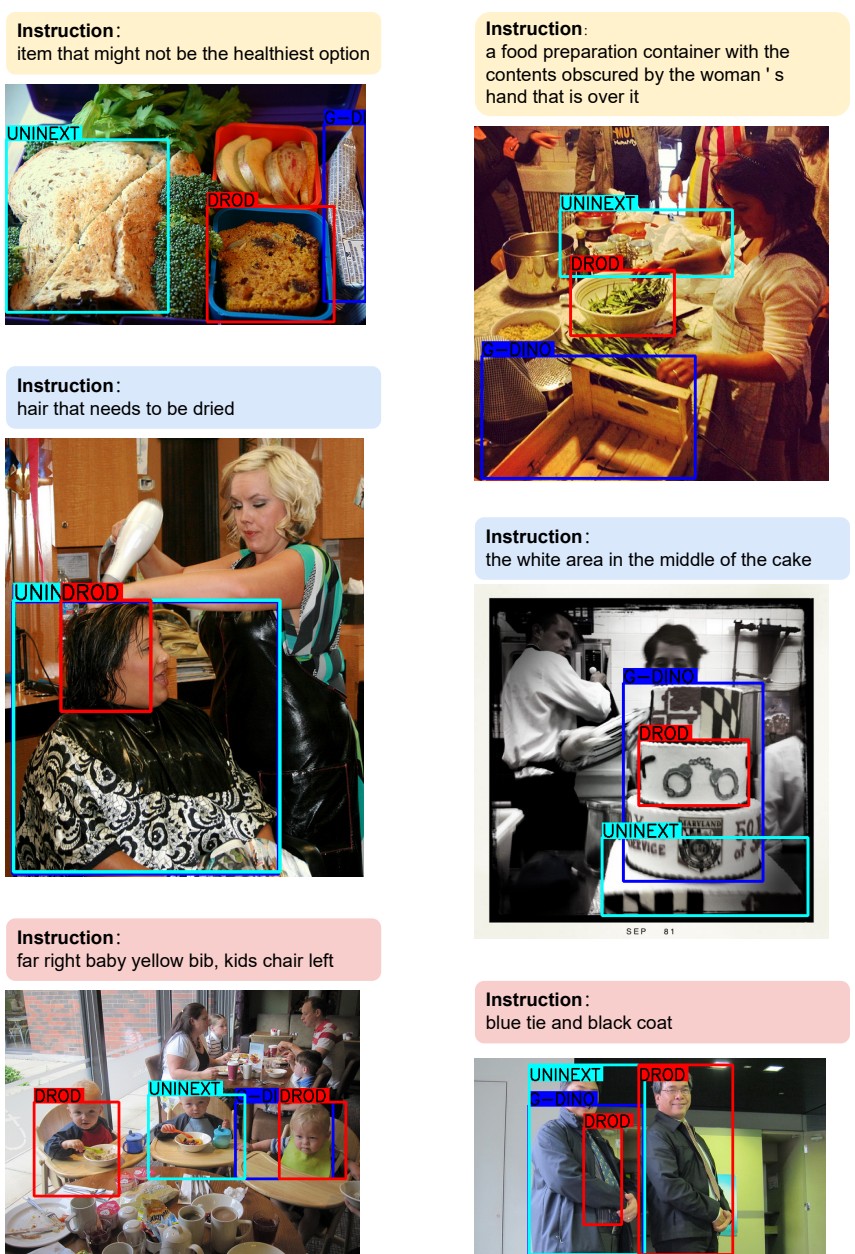

Figure 9: Visual comparison results. Our DROD well executes detection instruction compared to Grounding-DINO and UNINEXT.

# B EXPRESSION GENERATION VIA LOCAL PROMPT PIPELINE

## B.1 PROMPT AND GENERATION DISPLAY

We show the text prompts we use below in Table 4. For the LLaVA input, We randomly select one text prompt together with one image marked with object bbox. The inputs and samples generated from LLaVA are shown in Fig. 10.

- "Describe the objects in the red box."
- "Take a look at this image and describe What's in the red box."
- "Please provide a description of the object in the red box."
- "Could you describe the contents in the red box of the image for me?"
- "Use one sentence to index the objects in the red box."
- "Output a sentence describing the objects in the red box, so that people can locate the objects without ambiguity through this sentence."
- "Look carefully at the objects in the red box and describe them in one sentence to distinguish them from other objects."

Table 4: Text prompts we use for LLaVA inputs.

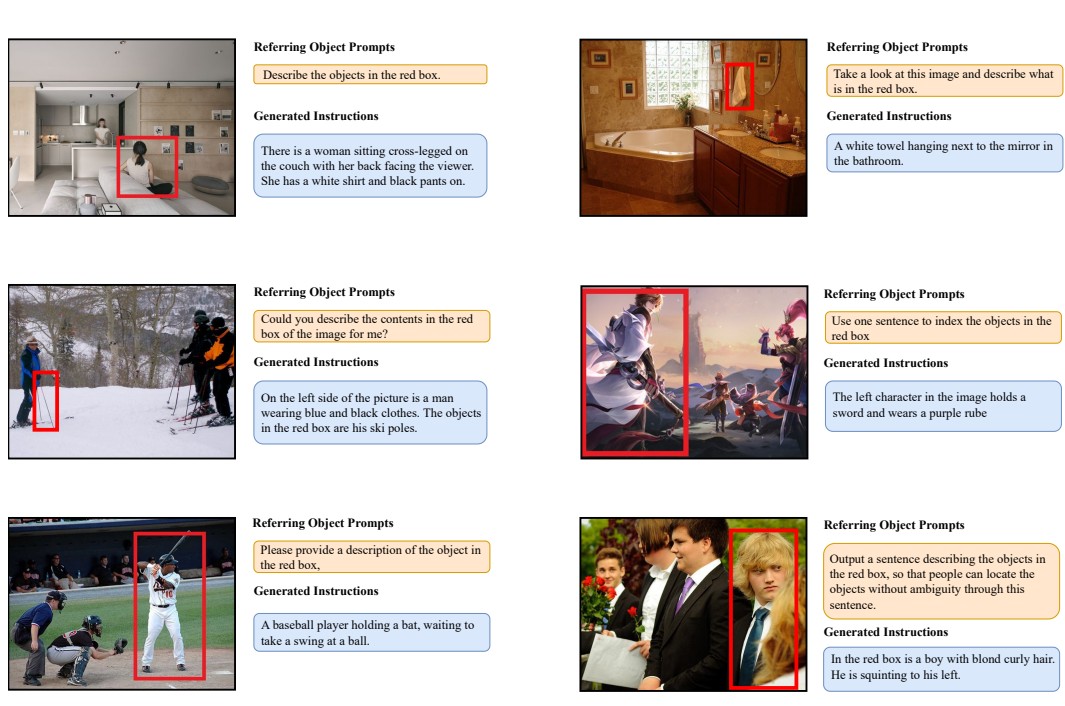

Figure 10: Examples of text prompts, images with marked object bboxs, and LLaVA outputs.

## B.2 FINE-TUNING SETTINGS

**Dataset Construction** We employed the entire dataset from RefCOCO/+/g for our fine-tuning endeavors, which primarily comprises triplets of images, bounding boxes, and instructions. The detection boxes are overlaid onto the images, serving as the visual input for LLAVA, while the instructions from the dataset are used as supervised outputs. The prompts are randomly selected from Table 4. The overall dataset consists of 386k instructions paired with 66k images. We chose not to utilize additional REC datasets for fine-tuning, as our fine-tuning objective is solely directed

towards enabling LLAVA to comprehend our task and acquire the ability to describe targets within red detection boxes.

**Training Process** Firstly, it is crucial to emphasize that in this paper, LLAVA does not refer to the model presented in the original LLAVA paper. Instead, it broadly denotes a multimodal paradigm where image information is abstracted into several tokens and inputted into a Large Language Model (LLM) for cross-modal attention with textual tokens. Our LLAVA architecture utilizes Q-Former to extract 32 tokens from the VIT features of the image, and these 32 tokens are subsequently mapped to the same feature space as textual tokens through a linear layer. During fine-tuning, only the final mapping linear layer is trained. This fine-tuning approach enables the model to learn the salience of the target object within the red bounding box in the input image while ensuring that the overall model avoids catastrophic forgetting. We initialize our model with the weights of MiniGPT-4.

**Hyperparameter Configuration and Implementation Details** We employed Vicuna 13B as our Large Language Model (LLM). The batch size and number of epochs were set to 32 and 30, respectively. The initial learning rate was set to $3 \times 10^{-5}$. Model training was conducted using 4 threads on a single A100 80GB GPU.

### B.3 CHAIN-OF-THOUGHT AND REFLECTION

After fine-tuning, the LLAVA model still exhibits a certain degree of hallucinations, manifesting as the generation of imaginary objects. Taking inspiration from the concept of thought chains in large language models, we manually constructed a thought chain specifically tailored for the task of describing target objects. In Table 5, we illustrate the prompts at each step of the thought chain: 1) Inquire about the category of the target object. 2) Inquire about the attributes possessed by the target object. 3) Inquire about the objects surrounding the target object. 4) Inquire about the relationship between the target object and its surrounding objects. Through these four steps, we can attain a comprehensive understanding of LLAVA's perception of the target object. However, hallucinations may still persist in the thought chain; therefore, in step 5), we prompt the model to reexamine the image and correct any errors. Finally, in step 6), we prompt LLAVA to produce the ultimate description of the target object.

1) "What is the object inside the red bounding box?"

2) "What are the attributes of the object within the red bounding box?"

3) "Which objects are around the target object in the red box?"

4) "What is the relationship between the object inside the red bounding box and the surrounding objects?"

5) "Please review the image once again, and if there are any inaccuracies in your previous answers regarding the object's attributes and relationships, kindly correct them."

6) "Look carefully at the objects in the red box and describe them in one sentence to distinguish them from other objects."

Table 5: Text prompts for guiding chain-of-thought and reflection.

## C    Expression Generation via GLOBAL PROMPT PIPELINE

We choose an image from Objects365 as an example to illustrate the instruction generation pipeline via our global prompt pipeline. This pipeline consists of inputs, two steps, in-context samples, and LLaMA outputs, which are illustrated one-by-one in the following:

**Inputs**    Our input is an example image with object bbox coordinates, which is shown in Fig. 11.

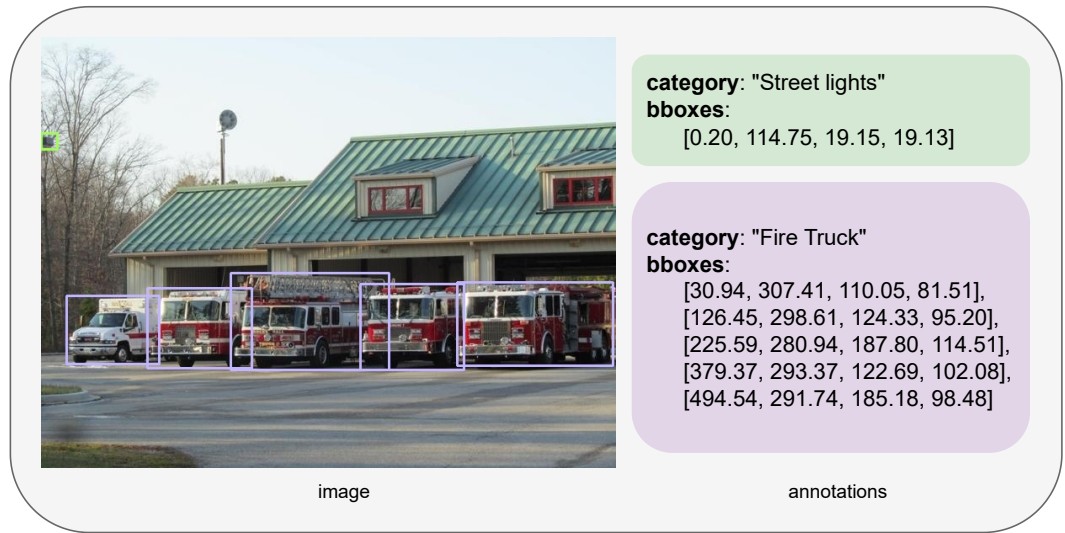

Figure 11: An input example. The left part is an image from the Objects365 dataset. The right part is the object category names and corresponding bounding boxes coordinates.

**Output**    We send the final text prompt shown in Table 9 to LLaMA. Table 6 shows the output expression of the inputs shown in Fig. 11. The detailed steps to prepare text prompts are illustrated as follows:

> **[Fire Truck]**
> (1) vehicle, emergency vehicle, fire engine, parked outside the fire station, an essential part of the fire station's resources, essential part of the fire station's resources
> (2) lined up in a neat row, ready for use, object parked in the row with other fire trucks, object with ladders and equipment
> **[Street Lights]**
> (1) light fixtures, outdoor lighting, two lights visible, providing illumination, source of illumination
> (2) objects providing illumination, objects in the parking lot, objects providing a clear representation of the overall setting, objects providing light for the parking lot

Table 6: LLaMA output for the inputs shown in Fig. 11.

Figure 11 and Table 6 depict the original input and final output of the global prompt pipeline, with the intermediary processing steps outlined below. Initially, the generation of the desired caption description for images lacking dense captions is required (**Step 1**). Subsequently, with the image fully textualized, we employ a detailed prompt design and seed example design to instruct LLAMA in outputting instructions describing the target object (**Step 2**).

**Step 1** We use LLaVA to generate image captions. Table 7 shows the text prompt list we use. One random text prompt from this list is combined with object content description (i.e., referring expression or category name) for LLaVA input. For each image, we send these prompts into LLaVA twice and obtain two diverse text descriptions. These two text descriptions are both utilized in the following steps. Examples of generated text descriptions are shown in Table 8.

- "Describe the following image in detail"
- "Provide a detailed description of the given image"
- "Give an elaborate explanation of the image you see"
- "Share a comprehensive rundown of the presented image"
- "Offer a thorough analysis of the image"
- "Explain the various aspects of the image before you",
- "Clarify the contents of the displayed image with great detail"
- "Characterize the image using a well-detailed description"
- "Break down the elements of the image in a detailed manner"
- "Walk through the important details of the image"
- "Portray the image with a rich, descriptive narrative"
- "Narrate the contents of the image with precision"
- "Analyze the image in a comprehensive and detailed manner"
- "Illustrate the image through a descriptive explanation"
- "Examine the image closely and share its details"
- "Write an exhaustive depiction of the given image"

Table 7: Text prompts we use for LLaVA. These prompts follow the author usage in the original LLaVA work.

**Prompt for LLaVA**
Provide a detailed description of the given image, including objects: *Street lights*, *Fire Truck*

**Output from LLaVA**
Output 1st The image displays a parking lot outside a fire station where several fire trucks are parked. In total, there are five fire trucks of varying sizes, all lined up neatly in the lot. Additionally, there are two street lights visible in the scene, providing illumination for the area. The fire trucks appear ready for use and are an bessential part of the fire station's resources.
Output 2nd The image depicts a parking lot outside a fire station, where several fire trucks are parked in a neat row. There are a total of five fire trucks of varying sizes, all aligned and ready for use. In addition to the fire trucks, there are two street lights visible in the scene, providing illumination for the parking lot. The overall setting gives a clear representation of the organized and prepared nature of the fire station.

Table 8: Generated examples of LLaVA based on one random text prompt from Table 7.

**Step 2** We write text prompt for LLaMA. In this step, the text prompt contains three parts including task descriptions, in-context examples, and text description of image (i.e., Table 8). An example text prompt for LLaMA is shown in Table 9 where an in-context sample is shown in Table 10.

```
task_description = f"""
```
*# Task description prompt*

## Establishing fundamental roles and task localization for LLM.
You are an AI visual assistant that can analyze a single image.
## Elucidating the form and significance of input information.
User will give you several sentences, describing the same image you are observing. In addition, specific interested object locations within the image are given, along with detailed coordinates. These coordinates are in the form of bounding boxes, represented as (x1, y1, x2, y2) with floating numbers ranging from 0 to 1. These values correspond to the left top x, left top y, right bottom x, and right bottom y.
## Explicating the output content and the encapsulated information.
Using the provided caption and bounding box information, give descriptions about the visual content of each interested objects as if you are seeing the image, as an assistant:

(1) give descriptions about the object itself, including **object types, object functions, object counts, object locations, object actions**, etc.
(2) give descriptions about the object and other objects, including **the relative positions between objects, the interaction between objects in the image**, etc.
## Emphasizing common output issues.
Descriptions should be a series of phrases, not whole sentence. Give descriptions for specific interested objects only, do not centered on other objects.
Again, give descriptions centered on specific objects only.

```
"""
```

```
image2text = f"""
```
*# Image description of LLaVA, the following two paragraphs are from Table 8*

The image displays a parking lot outside a fire station where several fire trucks are parked. In total, there are five fire trucks of varying sizes, all lined up neatly in the lot. Additionally, there are two street lights visible in the scene, providing illumination for the area. The fire trucks appear ready for use and are an bessential part of the fire station's resources.

The image depicts a parking lot outside a fire station, where several fire trucks are parked in a neat row. There are a total of five fire trucks of varying sizes, all aligned and ready for use. In addition to the fire trucks, there are two street lights visible in the scene, providing illumination for the parking lot. The overall setting gives a clear representation of the organized and prepared nature of the fire station.

Street lights: [0.0, 0.23, 0.03, 0.26]
Fire truck: [0.05, 0.6, 0.21, 0.76], [0.19, 0.58, 0.37, 0.77], [0.33, 0.55, 0.61, 0.77], [0.56, 0.57, 0.74, 0.77], [0.72, 0.57, 1.0, 0.76]
```
"""
```
*# Python code together with above text prompts are directly sent to LLaMA*

```python
messages = [{"role": system, "content": task_description}]
for in_context_example in in_context_examples:
    messages.append({"role": user, "content": in_context_example[
    "content"]})
    messages.append({"role": assistant, "content":
    in_context_example["response"]}
messages.append({"role": user, "content": image2text})
```

Table 9: An example of prompt generation for LLaMA input. For each image, we first obtain text description in Table 8. Then, we organize the above prompts and python code, together with in-context examples for LLaMA input. An in-context example is in Table 10.

**Example image2text**

Two children, a girl and a boy are practicing their writing.
Two children sit on a small seesaw in the sand.
Two children sitting on a teeter totter.
2 kids playing on a seesaw.
Two kids sit on a seesaw.
2 kids/two kids/two children: [0.09, 0.59, 0.49, 0.94]
girl: [0.09, 0.59, 0.23, 0.92]
boy: [0.34, 0.62, 0.49, 0.94]
seesaw/small seesaw/teeter totter: [0.06, 0.83, 0.57, 0.93]
sand: [0.01, 0.61, 1.0, 1.0]

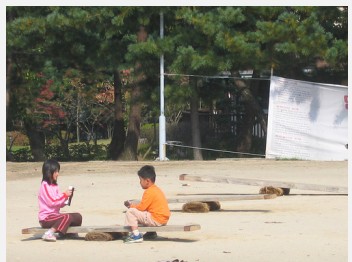

---

**Example Response**

**[2 kids/two kids/two children]**
(1) persons, children, two kids, children learning, kids playing, kids sitting, children practicing their writing
(2) 2 kids playing on a seesaw, Two kids sitting on a seesaw, a girl and a boy, two person on the teeter totter, kids by the sea
**[girl]**
(1) girl, kid on the left of the image, girl sitting, girl playing
(2) kid playing with the boy, girl sitting on the small seesaw, girl playing in the sand, girl reading a book, a friend of the boy on the right, the taller kid
**[boy]**
(1) boy, kid on the right, boy sitting, boy playing, boy practicing his writing
(2) kid playing with the girl, boy sitting on a teeter totter, a friend of the girl on the left, boy playing in the sand
**[seesaw/small seesaw/teeter totter]**
(1) small seesaw, teeter totter, item to be played on, common facilities in parks and playground, game of two people
(2) seesaw in the sand, item the kids are sitting on, item the girl is sitting on, item the boy is playing on
**[sand]**
(1) common by the sea, the background of the scene
(2) item on which the seesaw is placed, item on which the kids are standing

Table 10: An in-context example. We manually prepare three in-context examples and show one here. On the top block there are image captions and object bbox coordinates. Note that the displayed image is only for image caption reference and is not used in practice. The bottom block shows our expected expressions for each object. The expressions listed in (1) focus on the object attribute itself, and listed in (2) focus on relationship between current object and other objects in this image.

The in-context example in Table 10 follows the task description in Table 9 and primarily serves to standardize the output format of LLAMA. The desired output format is intended to resemble the **Example Response** presented in Table 9. [name] denotes the target currently being described, where the descriptions in (1) are relatively simple, pertaining only to the inherent attributes, and those in (2) entail more complex descriptions involving surrounding objects. This facilitates accurate correspondence between each description and the object it pertains to when parsing the output descriptions.

It is noteworthy that the prompt design is manually crafted. It draws significant inspiration from the prompt framework used by LLAVA, with specific modifications and supplements tailored to the requirements of our task. The entire prompt design underwent numerous iterations based on the output results, ensuring the stability of the output format and style.

# D VISUAL PROMPTING AND VISUAL-TEXTUAL MATCHING

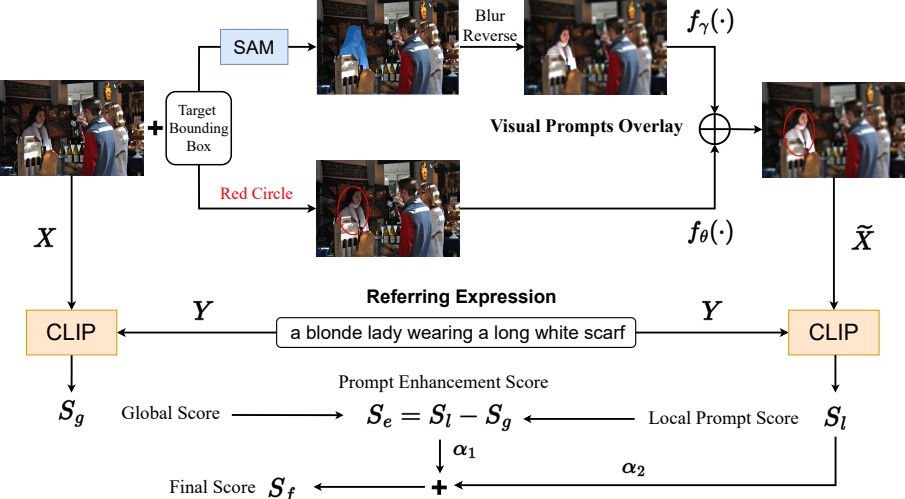

Figure 12: A detailed pipeline of visual prompting and visual-textual matching. We use visual prompting to emphasize target object in the image. Then, we use the CLIP model to compute the scores for expression filtering

.

Motivated by recent applications (Yang et al., 2023; Shtedritski et al., 2023) of visual language pretraining (VLP) models in solving zero-shot REC tasks, we discover that coupling VLP with visual prompting enables robust pairing of image regions and text on the basis of potent generalization. The efficacy of pairing is largely influenced by the configuration of visual prompting, which can be divided into two types as follows:

(i) Shape division of emphasized areas: box, circle and contour.

(ii) Highlighting methods for emphasized areas: crop-based, line-based, mask-based, grayscale reversion and blur reverion based image editing operations.

As shown in Fig. 12, after various combinations of experiments, we find the superposition of line-based circle prompt and blur reverse contour prompt yields the optimal visual prompting strategy. The image $\widetilde{X}$ after visual prompting can be expressed as:

$$\widetilde{X} = f_\theta(f_\gamma(X)) \quad \text{with} \quad f_\gamma(\cdot) = BR(\cdot, SAM(\cdot, B)) \quad f_\theta(\cdot) = RC(\cdot, B) \qquad (2)$$

where $B$ represents the box coordinates of the target, $BR$ and $RC$ represent the reverse Gaussian blur for the mask area and the inscribed ellipse for the target bounding box, respectively. $SAM$ denotes Segment Anything Model (Kirillov et al., 2023), which makes visual prompting more refined.

Once we obtain image $x$ and visually prompted $\widetilde{X}$, we send both of them to the CLIP model together with the generated expressions $Y$ from global prompt and local prompt pipelines. Then, we can compute the global score $S_g$ and local score $S_l$, and thus obtain the final score $S_f$ as illustrated in equation 1 for expression filtering.

# E    INSTRUCTION GENERATION FOR MINING CLUSTERS OF MULTI-OBJECTS

In order to find and summarize the common attributes among multiple objects in the clusters from DBSCAN (Ester et al., 1996), we use LLaMA for a further analysis. Table 11 shows the text prompt of task description and in-context examples we use for the LLaMA inputs. Then LLaMA produces expressions for multi-objects as shown in Fig. 4.

---

**Task Description**
You are an AI language assistant that can analyze phrases and sentenses. User will give you descriptions of several objects in an image. Descriptions of each object are several phrases or short sentences.

The given objects are expected to have similar properties. Based on the descriptions, find the common properties between given objects and summerize precisely as an assistant: common properties between objects can include same types, same functions, same color components, same poses, same relationships with other objects, engaging in the same activity, etc.

If there are no common properties between given objects, just tell that there are no common properties. Your summery should also be phrases. Do not repeat.

Give similarity between all given objects, contrary properties like different positions or different colors of clothes should not be included together in your descriptions.

**One In-context Example**
Prompt: Objects and their descriptions:
## object 2: girl sitting on bed, girl with toy, girl sitting on bed
## object 3: man looking down, boy sitting on the bed, man sitting on bed
Please find an summarize the similar properties of given objects.
Response: Summary of common properties of given objects:
## people on bed; person sitting on bed; people playing on bed; who sitting on bed;

---

Table 11: Task description and an in-context example for multi-objects instruction generation.

# F INSTRUCTION GROUPING

We use LLaMA to analyze different extents of description based on object category, attribute, and relations. For each instruction of single object in the InDET dataset, we use LLaMA to assign it into one of the preset 4 groups. Table 12 shows the text prompts of task description and in-context examples we use for instruction grouping. For instructions of multiple objects, we assign them to G5 if there is the combination (e.g., "and") of single instructions, or we assign them to G6 if the instructions are generated without concatenation.

---

**Task Description**
You are an AI language assistant that can analyze the language complexity of sentences or phrases.
User will give you a phrase or sentence describing a specific object in a image, which could be composed of nouns, adjectives, verbs, etc.
Grade the description according to its language complexity as an AI language assistant.

The language complexity of a phrase or sentence describing a specific object in an image can be graded into four levels:
**Level 0.** A single noun is used to give the object's name or type.
**Level 1.** A phrase with one or more nouns, verbs and abjectives is used to describe simple attributes of the object itself, such as its color, its function or purpose, location in the image, or actions.
**Level 2** A phrase with one or more nouns, verbs and abjectives is used to describe the object by referring to other objects in the image and describing their relationship, such as their relative positions or interactions.
**Level 3.** A long phrase or sentence is used to describe attributes of the object and also refer to a few other objects in detail, or describe a complicated or comprehensive/implicit relationship between multiple objects.
The level of descriptions increase as the language complexity and length increase, and also increase as the phrases or sentences become more descriptive and use more abjectives and nouns to describe the object.

**One In-context Example**
Prompt: Grade description: people who are sitting under an umbrella.
Response: My grading for description people who are sitting under an umbrella: This phrase is referring to the object of people, and gives simple object action of sitting and object relationship with the umbrella. The level of this description is: level 2.

---

Table 12: Task description and in-context examples for single-object instruction grouping.

# G MODEL AND IMPLEMENTATION DETAILS

In this section, we illustrate our model architecture and training configurations.

## G.1 MODEL ARCHITECTURE

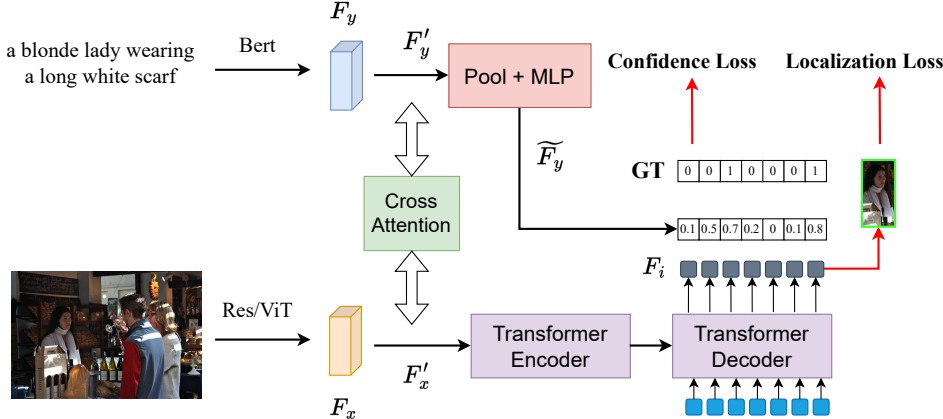

Figure 13: A detailed overview of our diversified referring object detection (DROD) model.

In the main text, We briefly introduced the DROD model architecture we use. Here, we introduce the overall model structure in Fig. 13.

**Visual Textual Feature Fusion** We use a general text encoder (Devlin et al., 2018) and vision encoder (Dosovitskiy et al., 2021) to obtain text features $F_y$ and image features $F_x$. In order to improve the attention of visual contents on the described regions by the text expressions, we conduct a multi-modality feature fusion. Specifically, we use a bidirectional cross attention (Bi-XAtt) module to enhance image and text features through information transfer between modalities. The enhanced image and text features are then added to the original features. This process can be formulated as follows:

$$F'_x = F_x + F_{y2x} \; ; \; F'_y = F_y + F_{x2y} \quad \text{with} \quad F_{y2x}, F_{x2y} = \text{Bi-XAtt}(F_x, F_y) \tag{3}$$

**Target Discovery and Retrieval** With the enhanced visual and language representations, we need to extract the targets referred to by the expression from the image features. Our DROD model applies the encoder-decoder architecture of Deformable DETR (Zhu et al., 2020), which allows for more flexible query retrieving. Here we provide a detailed introduction to our internal module.

The transformer encoder receives multi-scale visual features after text enhancement. Multi-scale Deformable encoder utilizes flexible attention at different scales and spatial shapes to obtain hierarchical characteristics for instances. In addition, following the design of two-stage Deformable DETR, we add an auxiliary prediction head for reference points. The top $N$ points are input into the decoder as the position prior.

The Transformer decoder uses learnable queries to retrieve instance-related information from encoded multi-scale visual features. The design of the query is critical for ensuring stability and efficiency in training, based on previous works (Meinhardt et al., 2022; Wang et al., 2021). The $N$ reference points served by encoder act as the position priors of the $N$ queries. The content part of each query is a static learnable vector. Moreover, following DINO, we add denoising queries to make the decoder's convergence more stable and faster. Through Deformable attention, $N$ queries efficiently retrieve instance embedding $F_i \in \mathbb{R}^{N \times d}$ from expression-aware visual features.

Finally, we need to select the instances referred to by expression from the $N$ instance proposals. As shown in Fig. 13, we use global average pooling and MLP to map the visual-aware expression feature $F'_y$ to the semantic space of the query embedding to obtain $\widetilde{F_y} \in \mathbb{R}^{1 \times d}$. Thus, the matching

score between each query proposal and the expression can be expressed by the cosine similarity between vectors. $S = F_i \times \widetilde{F_y}^\top$. In the inference stage, proposals with scores above threshold $\theta$ are selected. This flexible selecting strategy allows our architecture to output any number of results that satisfy user requirements. In the training stage, the overall model is supervised by a combination of confidence loss and localization loss.

## G.2 TRAINING CONFIGURATIONS

**Generation Engine** $\alpha_1$ in the expression filter responsible for balancing referentiality and semantic accuracy is set to 0.5. While generating multi-target expressions, DBSCAN clustering method has two hyperparameters: neighbourhood radius $eps$ is 1.5 and minimum neighbors of the core point $minPts$ is 2. The temperatures of LLaMA are set to 0.7.

**DROD Model** We use ResNet-50 (He et al., 2016) and ViT-Huge (Dosovitskiy et al., 2021) as visual encoders and Bert (Devlin et al., 2018) as the text encoder. The transformer encoder-decoder architecture consists of a six-layer encoder and a six-layer decoder. The number of object queries $N$ is set to 900. Our DROD model is initialized by weights pretrained on Objects365 released by UNINEXT (Yan et al., 2023). The optimizer we use is AdamW Loshchilov & Hutter (2019) with a learning rate of 2e-4, a weight decay of 0.05 and the warm-up steps are 400 with an initial learning rate of 4e-5. The model is trained on 32 and 16 V100 GPUs for pretraining and finetuning, respectively.

# H    ABLATION STUDIES

In this section, we study the necessity of finetuning LLaVA during expression generation from local prompt pipeline. Then, we investigate how different visual prompting strategies affect the CLIP model to maintain our expected expressions.

**LLaVA Finetuning**    In our InstructDET, we do not finetune LLaMA in global prompt pipeline, but finetune LLaVA in local prompt pipeline. There are two main reasons illustrated as follows:

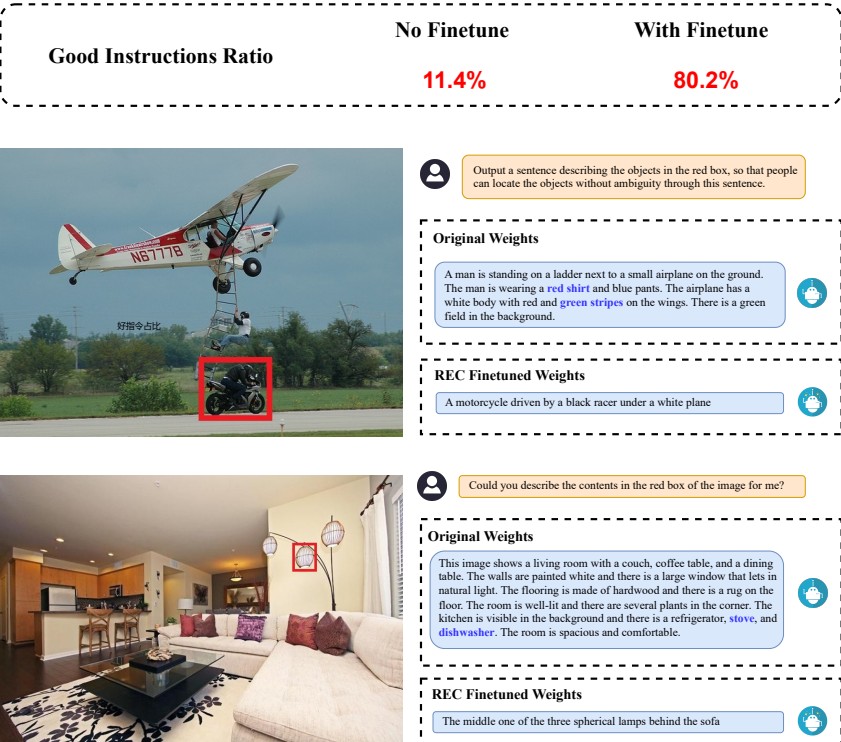

Figure 14: Comparison of expressions generated by LLaVA before and after our finetuning. Blue words indicate incorrect descriptions.

(i) Asking questions directly to LLaVA will get the answers with dense description, rather than the exact expressions we expect for the specific target objects.

(ii) Due to the limited number of visual tokens output by Q-Former, it is difficult to fully display all the visual features of the entire image. With limited visual information, LLaVA produces lengthy descriptions and thus lead to massive hallucinations (blue words in Fig. 14). The generated expressions in this way becomes unstable.

Based on above analysis, we partially finetune LLaVA on the linear layer that that transforms visual features into the semantic text embedding space. To this end, the model is updated to understand which area we aim to emphasize (i.e., marked in the red box in Fig. 14) and which visual features are important for target object description. Fig. 14 shows that after finetuning the linear layer using REC datasets, the probabilities of generating our expected high-quality expressions increases from 11.4% to 80.2%.

**Visual Prompting Selection**    The selection of visual prompting (VP) is crucial for expression filtering. We evaluate the affect of various visual prompting methods based on the retrieval performance of CLIP model in Table 13. First, we define a new metric called expression retrieval ratio, which indicates the proportion of correct expressions that can be retrieved based on the visual-textual matching of CLIP model. During testing, each minibatch contains $k$ correct target expressions, and the remaining expressions are negative samples. We take the expressions with the top $k$ Clip score as

Table 13: Visual Prompting (VP) Evaluation. Different VP methods make differences in the ability of CLIP to retrieve high-quality expressions.

| Strategy | Visual Prompting | | Expression Retrieval (%) | |
|---|---|---|---|---|
| | Shape | Tool | Easy ↑ | Hard ↑ |
| 1 | | Crop | 31.52 | 24.09 |
| 2 | | Gray | 39.66 | 31.85 |
| 3 | Box | Line | 50.12 | 42.30 |
| 4 | | Mask | 48.29 | 39.93 |
| 5 | | Blur | 52.75 | 44.30 |
| 6 | | Line | 52.78 | 44.33 |
| 7 | Circle | Mask | 51.01 | 42.87 |
| 8 | | Blur | 54.13 | 46.22 |
| 9 | | Line | 52.34 | 44.05 |
| 10 | Contour | Mask | 51.90 | 43.44 |
| 11 | | Blur | 55.79 | 47.78 |
| 12 | VP3 + VP11 | | 56.81 | 48.97 |
| 13 | **VP6 + VP11** | | **58.29** | **50.99** |

the retrieved expressions. Finally, the proportion of correct expressions in the retrieved expressions is the expression retrieval ratio. Under the Easy setting, the negative samples in the minibatch come from other images. Under the Hard setting, the negative samples in the minibatch may come from different targets in the same image. Table 13 shows that cropping (1) and grayscale reversion (2) methods achieve poor results. Because cropping loses all surrounding environment information, and grayscale reversion loses all color features. The best single prompt is the contour blur reversion. The best combination prompt is the contour blur reversion and circle line. Circle line can indicate the rough areas that needs attention, and contour blur reversion can highlight the target object in a fine-grained manner that eliminates background interference. Inevitably, the prior knowledge in the CLIP model is also important to the difference in visual prompt effects. In CLIP's pre-trained web-scale dataset, images with red circles often indicate that the targets in the red circles are more important and need to be noticed. A large amount of photography also exists in the web-scale dataset, which employs "Bokeh" to blur the background and highlight the subject.

# I SYNONYMOUS REWRITE

In post processing, we utilize LLaMA to do synonymous rewriting to further diversify the generated expressions. The prompt we used is shown in Table 14.

> **Synonymous Rewriting Prompt**
> I want you to act as a synonymous expression provider. I will give you a text of phrase or short sentence, which is an expression that describes a main object while mentioning some other objects. And you will reply to me with a new expression that have the same semantic meaning and describe the same main object as the provided expression. The new expressions should also be phrases or short sentences no longer than 25 words. Do not write explanations on replies. Do not repeat.

Table 14: Synonymous Rewriting Prompt for LLaMA.

# J SUPPLEMENT COMPARISONS

## J.1 EVALUATION ON INDET

Table 15: Evaluation results on our InDET. We show the object bbox average precision (AP) values (%).

| Method | Backbone | AP | AP by Group | | | | | |
|---|---|---|---|---|---|---|---|---|
| | | | G1 | G2 | G3 | G4 | G5 | G6 |
| MDETR | ResNet101 | 34.86 | 47.44 | 46.79 | 34.14 | 23.22 | 25.91 | 28.17 |
| G-DINO | SwinB | 35.96 | 47.10 | 47.17 | 35.29 | 26.84 | 27.95 | 27.61 |
| UNINEXT | ResNet50 | 43.37 | 54.49 | 54.52 | 44.49 | 37.17 | 31.41 | 32.01 |
| DROD (Ours)[1] | ResNet50 | 62.24 | 67.14 | 67.34 | 60.89 | 55.10 | 70.15 | 74.22 |
| DROD (Ours)[2] | ResNet50 | 62.34 | 67.22 | 68.04 | 61.09 | 55.42 | 68.60 | 72.91 |
| DROD (Ours) | ViT-H | 66.90 | 72.53 | 72.47 | 66.42 | 59.86 | 73.34 | 75.92 |

[1] For fair comparison, our DROD model in Table 2 only utilizes RefCOCO/+/g datasets but with diversified instructions, which is partial of InDET.
[2] Here we add evaluation results of DROD model trained on full InDET.

## J.2 EVALUATION ON REFCOCO/G/+

Table 16: Supplementary Evaluation results on the RefCOCO/g/+ datasets. We follow evaluation protocols to report AP values (%) of comparing methods. We use the notations "CC", "VG", "OI", "O365", "RIGame", for COCO, Visual Genome, OpenImage, Objects365, ReferItGame, respectively.

| Method | Backbone | Data | RefCOCO | | | RefCOCO+ | | | RefCOCOg | |
|---|---|---|---|---|---|---|---|---|---|---|
| | | | val | testA | testB | val | testA | testB | val-u | test-u |
| RefTR | ResNet101 | VG | 85.65 | 88.73 | 81.16 | 77.55 | 82.26 | 68.99 | 79.25 | 80.01 |
| SeqTR | DarkNet53 | VG,RIGame,Flickr,RefC | 87.00 | 90.15 | 83.59 | 78.69 | 84.51 | 71.87 | 82.69 | 83.37 |
| MDETR | ResNet101 | GoldG,CC,RefC | 86.75 | 89.58 | 81.41 | 79.52 | 84.09 | 70.62 | 81.64 | 80.89 |
| G-DINO | SwinB | O365,CC,RefC,GoldG,etc | 83.95 | 87.79 | 79.16 | 72.91 | 80.91 | 62.96 | 76.98 | 76.76 |
| PolyFormer | SwinL | GoldG,ReferIt,RefC | 90.38 | 92.89 | 87.16 | 84.98 | 89.77 | 77.97 | 85.83 | 85.91 |
| UNINEXT | ResNet50 | O365,CC,RefC | 87.64 | 90.35 | 83.49 | 78.14 | 83.22 | 68.71 | 80.96 | 81.86 |
| DROD (Ours)[1] | ResNet50 | O365,CC,InDET | 88.92 | 90.86 | 85.57 | 78.27 | 83.39 | 71.04 | 83.01 | 82.91 |
| DROD (Ours)[2] | ResNet50 | O365,CC,InDET | 89.85 | 92.03 | 87.24 | 80.50 | 85.87 | 73.61 | 83.93 | 84.73 |
| DROD (Ours)[3] | ViT-H | O365,CC,InDET | 92.93 | 94.47 | 91.13 | 86.20 | 89.82 | 80.86 | 88.62 | 89.46 |

[1] For fair comparison, our DROD model in Table 3 only utilizes RefCOCO/+/g datasets but with diversified instructions, which is a part of InDET.
[2] Here we add evaluation results of DROD model trained on full InDET.
[3] DROD here with ViT-H as backbone also utilizes RefCOCO/+/g datasets with diversified instructions.

We have supplemented the comparisons with more specialized models, such as Polyformer(Liu et al., 2023b). With comparable magnitudes in the backbone network, our model continues to exhibit advantages.

## K   TIME CONSUMPTION

Table 17: Time Consumption Statistics. We use "instr." as the short for "instruction".

| Pipeline | Step | Main Model | BatchSize | Time | Avg |
|---|---|---|---|---|---|
| Global Prompt | Image Caption | LLaVA | 4 | 10.71 s/batch | 2.68 s/image |
| | Instr. Generation | LLaMA | 1 | 17.25 s/image | 0.63 s/instr. |
| Local Prompt | Instr. Generation | LLaVA | 16 | 2.47 s/batch | 0.15 s/instr. |
| Post-Processing | Instr. filtration | CLIP | 1 | 0.186 s/image | 0.016 s/box |
| | Multi-Object Instr. | DBSCAN | 1 | 0.053 s/image | - |
| | Instr. Grouping | LLaMA | 1 | 1.43e-06 s/instr. | - |

Table 17 shows the time consumption of the main steps in our instruction generation procedure. The data is obtained from NVIDIA-v100-32G machines. Obviously, the most time consuming steps are those who depend on large models. Except for the limited hardware performance, another reason why the instruction generation step in global prompt pipeline is very time consuming is that we have very long LLaMA prompts which include three in-context examples and we also desire long response which could include more diversified instruction for each object in image. So this step requires multiple machines to generate simultaneously. While the text received and generated in each forward pass is relatively short in the local prompt pipeline, the generation efficiency is significantly constrained by the fact that only one instruction can be generated per forward pass. Therefore, we address this limitation by maximizing parallelism through the increase in batch size, allowing the simultaneous generation of multiple instructions.

