# OpenReview forum: "InstructDET: Diversifying Referring Object Detection with Generalized Instructions"
_ICLR.cc/2024/Conference — ICLR 2024 poster_

### Official Review · Reviewer_n3DU · 2023-10-30

**Soundness:** 3 good
**Presentation:** 2 fair
**Contribution:** 2 fair
**Rating:** 5
**Confidence:** 4

**Summary:**

This paper introduces a new grounding dataset, which is created with LLM (LLAMA) and VLM (LLAVA). They propose to leverage LLAMA in the single modality pathway, which receives the text only descriptions and bounding boxes to create annotations, in the same spirits to LLAVA. And LLAVA is leveraged in the multi modality pathway. After creating the dataset automatically, they leverage CLIP to drop out the annotations with low quality. A simple baseline model has been proposed to showcase the effectiveness.

**Strengths:**

The proposed dataset may be a good resource for future research in visual grounding tasks.

**Weaknesses:**

Weaknesses:
1. The major concern I have is regarding the quality dataset. The author uses the open-source model LLAMA, which is not instruction-tuned, in the single modality pathway. As opposed to GPT4 used in LLAVA’s instruction tuning dataset. In addition, LLAVA is used in multi-modality pathways. I have doubts about the quality of the dataset since both LLAVA and LLAMA have problems of their own, such as hallucinations and incapability to describe the objects of small scales.

2. The experiments are insufficient. The author only presents the result in their own proposed dataset and the original ref coco benchmark. I am not certain how much the dataset actually helps the performance of the model, or helps equip the model with new capabilities. Specifically, the performance on refcoco is not extraordinarily high and is not compared with SOTA methods such as poly former.

3. What does the author mean by “initializing LLAVA with minigpt4 weights”? I think LLAVA finetunes the entire LLM, while minigpt4 only finetunes a linear layer, and the architectures are different.

**Questions:**

See weakness

---

> ### Author Response · Authors · 2023-11-23
> **Response to Reviewer n3DU For Raised Weaknesses 1&2**
>
> Dear Reviewer  n3DU,
>
> We would like to thank you for providing valuable comments and we answer the raised questions as below. We have included all discussions and results in our revised manuscript (blue content).
>
> &nbsp;
>
> **Response for Weakness 1.**
>
> We apologize for the confusion regarding our dataset and we will clarify in the following. In the global prompt pipeline (i.e., single modality pathway), we leverage a variant model of LLaMA named as Vicuna 13B. The Vicuna model has gone through instruction tuning and thus is capable of following our instructions related to user expression generation. For the GPT-4 employment, we have found out that we need to pay for its API usage and have estimated the fee cost to produce the total amount of our required user expressions (i.e., around tens of millions). This cost has exceeded our budget. Nevertheless, we have evaluated on a toy dataset to validate that in our ROD scenario, the expressions generated by GPT-4 are not obviously advantageous compared to those from the Vicuna model.
>
> In our local prompt pipeline (i.e., multi-modality pathway), we observe that a direct expression generation by LLaVA introduces model hallucinations. To mitigate such limitation, we employ the following three steps. First, we finetune LLaVA on the REC dataset. This finetuning transfers the model attention from global images into local objects (as shown in Appendix B.2). Second, we leverage the chain-of-thought and reflection approaches (e.g., in-context learning) to prompt LLaVA to infer results via reasoning (as shown in Appendix B.3) . Third, we further filtered the generated expressions using a CLIP model (Section 3.3). We evaluate the quality of expressions generated in each step by manual verification. We randomly sample 500 expressions from each step and verify the amount ratio of human-like expressions. The results are shown in the following table where we can gradually improve the quality of generated expressions.
> | Stage | Ratio of human-like expressions |
> | :-----| :----: |
> | Raw LLaVA (Vicuna) | 11.4 % |
> | After Finetune | 80.2 % |
> | After CoT | 84.6 % |
> | After Filtering | 97.8 % |
>
> Revised Sections:
>
> (1) We add footnote 2 on the fourth page to specify the fine-tuned version of LLaMA that was utilized.
>
> (2) We provide detailed explanations in Appendix B.2 and B.3 regarding the methods employed to enhance the generation quality of LLaVA through fine-tuning and thought of chains.
>
> &nbsp;
>
> **Response for Weakness 2. The experiments are insufficient. The author only presents the result in their own proposed dataset and the original ref coco benchmark. I am not certain how much the dataset actually helps the performance of the model, or helps equip the model with new capabilities.**
>
> Thanks for your valuable comments. From Table 2, we observe that ROD models trained on the existing datasets are not effective to ground objects in our InDET test set. This is because our InDET dataset contains much more diversity of human-like expressions than existing datasets. To this end, the evaluation on the RefCOCO test set, whose diversity is limited, is not effective to demonstrate the superior model generalizations brought by our InDET dataset.
>
> In order to further evaluate how our dataset helps model to improve performance or equip new capabilities, we have performed a new evaluation below. We train existing models (i.e., MDETR and UNINEXT)  using different training sets (i.e., RefCOCO/+/g, Flicker, and our InDET). Then, we select 2000 test samples from our InDET test set where these samples focus on logical reasoning in ROD. The evaluation results are shown in the following table:
> | Method | Datasets | AP |
> | :-----| :----: |:----: |
> | MDETR | Refcoco/+/g | 18.91 |
> | MDETR | Flicker | 10.14 |
> | MDETR | InDET | 49.37 |
> | UNINEXT | Refcoco/+/g| 22.88 |
> | UNINEXT | Flicker | 16.07 |
> | UNINEXT | InDET | 56.90 |
>
> This table indicats that exisiting models achieve significant performance gain by using our InDET dataset. As our test set focuses on logic reasoning, we can conclude that our InDET improves the logic reasoning of representative existing models. We have added this comparison in Fig.7 together with visual examples.
>
> Revised Sections:
>
> In Sec. 6, we incorporate further discussions on multimodal logical reasoning capabilities. Additionally,  Fig. 7 provides a quantitative and qualitative analysis of this ability.

---

> ### Author Response · Authors · 2023-11-23
> **Response to Reviewer n3DU For Raised Weaknesses 2&3**
>
> **Response for Weakness 2. Specifically, the performance on refcoco is not extraordinarily high and is not compared with SOTA methods such as poly former.**
>
> Thanks for your comments. We would like to illustrate our initial motivation on showing Table 3. We aim to demonstrate that models trained on our InDET dataset do not decrease accuracies on existing test sets. We have checked PolyFormer and find it achieves a higher accuracy on RefCOCO. There are two reasons for this. First, our DROD model is a prevalent ROD model that is not specifically designed for a particular dataset, while PolyFormer is. The backbone of our DROD and PolyFormer differs significantly (i.e., ResNet 50 v.s. Swin-L) as well. Such difference naturally brings different abilities of feature representation. Second, we do not specifically tune hyperparameters of our DROD model for RefCOCO. This is because we do not focus on achieving accuracy gain on RefCOCO as much as possible. We only aim to show our DROD model performs well on existing ROD datasets. In contrast, we focus on demonstrating our model generalizations by evaluating on our InDET test set where our DROD model achieves higher accuracy than existing methods. Specifically, we have appended the results of Polyformer together and more of our DROD results in Appendix J.2. (blue content) in our revised manuscript.
>
> &nbsp;
>
> **Response for Weakness 3.**
>
> We apologize for the confusion and would like to make a clarification here. The LLaVA we mention here is indeed a multimodal paradigm that maps visual features into token embeddings with further alignment. This paradigm is similar to minigpt4 and differs from the structure from the original paper. We have changed this expression into "initializing a derived model from LLaVA with minigpt4 weights''.
>
> Revised Content: We have added the footnote on page 4 for clarification.

---

### Official Review · Reviewer_nJxy · 2023-11-01

**Soundness:** 2 fair
**Presentation:** 2 fair
**Contribution:** 2 fair
**Rating:** 6
**Confidence:** 3

**Summary:**

The paper introduces "InstructDET," a novel method for referring object detection (ROD) that identifies and localizes target objects in images using user instructions. This approach is an advancement from the traditional referring expressions (REC) and places emphasis on generating a wide variety of user-centric instructions linked to object detection tasks using VLM and LLMs
The InstructDET dataset is a compilation of images, their associated bounding boxes (bbxs), and the generalized instructions curated from VLM and LLM. Models trained on the InDET dataset demonstrated enhanced performance, outclassing existing methods on traditional REC datasets and the new InDET test set. InstructDET showcases the potential to expand by integrating any image accompanied by object bounding boxes.

**Strengths:**

1. The paper proposes a novel and interesting way to produce rich visual descriptions of objects and their relationships using LLM and VLM
2. I like the approach for removing hallucination from the VLM model using vision language matching for CLIP.
3. This approach is scaleable which means that the results can continue to improve with more bbox data.

**Weaknesses:**

1. It's not clear from the paper that the method comparison in table 2 is fair because it's not clear if the models were trained on the same images and bounding boxes
2. One additional interesting experiment that would make this a stronger paper - run SAM on web scale images to get bboxes for images and then scale up this approach further to understand if the benefits saturate

**Questions:**

Can you please address weakness 1?

---

> ### Author Response · Authors · 2023-11-23
> **Response to Reviewer nJxy For Raised Weaknesses and Questions**
>
> Dear Reviewer nJxy,
>
> We would like to thank you for providing valuable comments and we answer the raised questions as below. We have included all discussions and results in our revised manuscript (blue content).
>
> **Weakness 1. It's not clear from the paper that the method comparison in table 2 is fair because it's not clear if the models were trained on the same images and bounding boxes**
>
> We would like to clarify that in Table 2 where we evaluate on the InDET test set, existing methods (i.e., MDETR, G-DINO, UNINEXT) are trained on RefCOCO/+/g datasets where the images and bounding boxes are the same. For a fair comparison, our DROD model in Table 2 only utilizes RefCOCO/+/g datasets but with diversified instructions. Our superior results indicate that our diversified instructions enrich the alignment of visual objects and language expressions in DROD model.
>
> On the other hand, we perform another evaluation below where all the methods only use RefCOCO dataset for training and testing. The training data of existing methods is the RefCOCO training set, while the training data of our DROD is the same training set but with diversified instructions. For the testing data, all the methods use RefCOCO testing set but with diversified instructions. Through this configuration, we would like to show that existing models trained on RefCOCO dataset are not effective to generalize, while our diversified instructions improve our model generalizations to achieve favorable performance.
>
> | Method | Images (train and test) | AP |
> | :-----| :----: | :----: |
> | MDETR | Refcoco | 36.83 |
> | G-DINO | Refcoco | 37.90 |
> | UNINEXT | Refcoco | 44.70 |
> | DROD (Ours) | Refcoco | 64.38 |
>
> The results on the table show that the diversified instructions improve our model performance under the same image and bbx dataset.
> &nbsp;
>
> **Weakness 2. One additional interesting experiment that would make this a stronger paper - run SAM on web scale images to get bboxes for images and then scale up this approach further to understand if the benefits saturate**
>
> Thanks for your comments. Our data-centric method is capable of generating image-bbx-instruction triplets (i.e., expanding dataset) automatically given bbxs predicted by SAM on web scale images. And our ultimate goal is to train a ROD model with this dataset to generalize ROD on a wide range of images and user expressions. As the size of dataset increases, we need to redesign this ROD model from the perspectives of model size and training configurations accordingly. Such effort needs a significant amount of time and we are unable to accomplish it at present. Nevertheless, we will definitely set this direction as our future work and introduce possible solutions (e.g., advanced learning strategy such as curriculum learning) to overcome the saturation if it really happens.

---

### Official Review · Reviewer_uAXw · 2023-11-01

**Soundness:** 2 fair
**Presentation:** 2 fair
**Contribution:** 3 good
**Rating:** 6
**Confidence:** 3

**Summary:**

This paper introduces InstructDET as new method for referring object detection (ROD). The paper identifies the lack of diversity in the user prompt in existing ROD datasets as a critical limitations. To address this limitation, a data pipeline based on query of foundation models, such as LLaVA and LLaMA is built. This pipeline creates accurate and diverse object reference via a combination of prompt engineering and heuristics. It first create object and image level expressions via LLaVA, then reject hallucination via CLIP, and finally diversify the expressions via LLaMA. The resultant expressions can then serves as training data for the ROD task. The paper shows that the dataset curated from the proposed procedure can significantly improve the model performance.

**Strengths:**

* This paper provides an interesting research direction in referring object detection. While many existing works attempt to improve ROD via better model design, this work suggests that it could be fruitful to enrich existing dataset via a combination of data mining and/or prompt engineering from existing foundation models. It would be of great interests to the community in general.

* The method taken from this paper seems to quite novel. While the use of foundation models as a data curation tool is not new (as fully acknowledged in this work, and is the practice used in LLaVA which is in return used heavily in this work), this work has explored some novel ideas to improve this procedure. For example, the "vision and language mapping" via CLIP as a verification step is particularly interesting.

* The resultant model trained on the curated dataset seems to outperform prior arts by a large margin, even with the same network architecture.

**Weaknesses:**

* There are many language errors throughout the submission, including strange use of words, grammatical mistakes etc.. It is still possible to comprehend the ideas despite those writing issues, but this submission can use additional proofreading. Part of the issue is with the naming of concepts. The "pathways" are easily confused with the terminology used to describe parallel branches in a deep network. The "dropout" can be confused with the regularization method. These names are unfortunate in the sense that they are in fact significant distractors for readers in this field, and not really descriptive of the concepts.

* I find the definition of "single" versus "multi" modality pathways confusing. To my best understanding, the "single modality pathways" refers to a sub-process in the data generation that uses LLaVA to create output texts descriptions for an input image. As such, it needs both an image and some text prompts to obtain text result. The "multi modality pathways" refers to a procedure that similarly use LLavA to create results from image & text prompts. The two pathways have the same form of inputs and outputs. Why one is "single modality", while the other is "multi modality"? It is also hard to see, at least from the discussions provided in the submission, why the "single modality" case can use an out-of-box LLaVA model, whereas the "multi-modality pathways" must use a finetuned version.

* While the main contribution from this paper is the proposed data generation step procedure, the details are not included in a way that would allow other researchers to reproduce the results. For example, Section 3.1 (combined with appedix C) describes the "single modality" pathway. However, critical details such as the procedure to generate prompts are left in a rather abstract level. I don't think it would be possible for reader to re-produce the method described. Similarly, in Section 3.2. the paper does not precisely describe the procedure. For example, while the paper mentions fine-tuning LLaVA on some REC dataset, it does not describe the exact dataset nor the exact training setting for this. Another example is in the "post-processing" section. It is not at all clear from reading the paper how the expressions are "verify and remove", and how LLaMA is used to "diversify" the expressions.

**Questions:**

* Could you include more details in terms of the prompts that are use, the exact procedure to generate them (number of queries, the services used, costs etc.)? I think these are interesting information for the community that should be included to the paper wherever appropriate.

* Do you plan to release the dataset curated for this paper?

---

> ### Author Response · Authors · 2023-11-23
> **Response to Reviewer uAXw For Raised Weaknesses 1&2**
>
> Dear Reviewer uAXw,
>
> We would like to thank you for providing valuable comments and we answer the raised questions as below. We have included all discussions results in our revised manuscript (blue content).
>
> **Response for Weakness 1.**
>
> We apologize for making you confused while presenting our method and would make a clarification here. The "pathway'' we use is to describe that in our framework, there are two pipelines to produce expressions (as shown in Fig. 2).  We have replaced "pathway'' with "pipeline'' in our revised manuscript. For "dropout'', we mean to wipe out expressions that are inaccurately generated via both LLaVA and LLaMA. We have replaced "dropout'' with "filter'' in our revised manuscript. Moreover, we have thoroughly proofread our paper to improve our presentation for reducing distractions of readers. For example, in Sec 3.1, we revised "The text prompt we use for LLaVA contains our language guidance **that the specific object category** shall be mentioned" into "The text prompt we use for LLaVA contains our language guidance **emphasizing that several specific interested object categories** shall be mentioned". And in Sec 3.3, we revised "The global image description is mitigated via our model finetuning to focus on local object, but not completely disappeared" into "The **tendency to generate** global image description is mitigated via our model finetuning to focus on local object, but not completely disappeared".
>
> &nbsp;
>
> **Response for Weakness 2.**
>
> We apologize for the confusion and make the clarification here. We agree that "single modality pipeline'' refers to the global image description that involves LLaVA for prediction. Specifically as illustrated in Sec. 3.1, we first obtain global image description and then sent this description, object bbxs axis, and in-context examples to LLaMA. Originally, we mean "single modality'' because we feel all the elements sent to LLaMA are text tokens. And we mean "multi-modality'' because we send visual images with marked bbxs, together with text tokens, to LLaVA as illustrated in Sec. 3.2.
>
> For your comments, we agree that it is inappropriate to name same form of inputs and outputs as "single modality'' and "multi modality''. To this end, we have modified our illustration that we replace "single modality pathway'' with "global prompt pipeline'', and replace "multi-modality pathway'' with "local prompt pipeline''. This replacement is based on the functions of LLaVA. In "global prompt pipeline'', we use LLaVA to obtain the global text description (i.e., Sec. 3.1). In "local prompt pipeline'', we use LLaVA directly to perceive the local visual objects (i.e., Sec. 3.2). We have revised these illustrations in our revised manuscript according to your advice.
>
> The reason why we use an out-of-box LLaVA model in "global prompt pipeline'' (i.e., "single modality'') is that LLaVA only provides global image description for an input image. In this pipeline, expressions are generated via LLaMA that takes only text tokens as inputs. As LLaVA is initially trained via image and text pairs, there is no need to finetune LLaVA to obtain global image descriptions. However, in the "local prompt pipeline'' (i.e., "multi modality''), we directly leverage LLaVA to generate expressions by taking images with marked bbxs and text tokens as ipnuts. As we expect LLaVA to effectively perceive local regions around each object, a finetuning is required to transfer the global image description of LLaVA into local perception.  In Figure 14 of Appendix H, we show that without fine-tuning, LLaVA achieves only 11.4% in generating local target descriptions. However, when fine-tuned on the REC dataset, LLaVA is improved up to 80.2% .

---

> ### Author Response · Authors · 2023-11-23
> **Response to Reviewer uAXw For Raised Weakness 3 and Questions 1&2**
>
> **Response for Weakness 3.**
>
> We apologize for the unclear presentation and would elucidate more data generation details below .
>
> For the procedure to generate prompts in Sec. 3.1 and appendix C, we have added more explanations in our revised manuscript to make a clearer illustration for prompt generation. The procedure of single modality pathway (i.e., global prompt pipeline) is designed to generate text instructions targeting for interested objects in an given image. The generation procedure can be summarized as follows,
>
> **Inputs.**  An image and it's bounding boxes of interested objects.
>
> **Step1.**  Obtaining global image description through LLAVA. Prompts for LLAVA to generate image descriptions are empirically designed and provided in Table 7.
>
> **Step2.** Generating text instructions corresponding to interested objects by LLAMA. Prompts for LLAMA consist of three parts: image information, task description, and in-context examples.
> - Image information: the global image description generated in step 1 and it's bounding boxes of interested objects.
> - Task description: describing the instruction generation task and emphasizing the outputs to LLAMA. The prompts are empirically designed and provided in Table 9.
> - In-context examples: designing three examples of input-output pair for LLAMA to elucidate the task clearer. These examples are also empirically designed and one of them is provided in Table 10.
>
> The task description and in-context examples are fixed and applied for all images. Only image description is generated according to different images. Furthermore, we will release the code of our generation procedure for public usage.
>
> &nbsp;
>
> For the LLaVA finetuning, we employ the entire dataset from RefCOCO/+/g for our fine-tuning endeavors, which primarily comprises triplets of images, bounding boxes, and instructions. The overall dataset consists of 386k instructions paired with 66k images. During fine-tuning, only the final mapping linear layer is trained. This fine-tuning approach enables the model to enrich the attention on the target object within the red bounding box in the input image while ensuring that the overall model avoids catastrophic forgetting. We initialize our model (i.e., a variant of LLaVA) with MiniGPT-4 weights. We employ Vicuna 13B as our Large Language Model (LLM). The batch size and number of epochs are set to 32 and 30, respectively. The initial learning rate is set as $3\times 10^{-5}$. Model training is conducted via 4 threads on a single A100 80GB GPU.  We have provided additional details and pertinent explanations in Appendix B.2 for reference.
>
> &nbsp;
>
> For the post-processing step, after the generation procedure, we have checked all the generated expressions and removed the the same expressions. We use LLaMA to further diversify the generated instructions by performing synonymous rewriting, the exact prompts we use is added to the Sec.I in our revised manuscript.
>
> We will release the related code for our implementation for further research.
>
> &nbsp;
>
> **Response for Question 1.**
>
> Sure, we'd love to share more details of the generation procedure in InDET.
>
> To be clear, the diversified instructions in InDET are generated by the procedure we illustrate in Sec. 3, which leverages LLAMA and LLAVA. The prompts for LLAMA and LLAVA are empirically designed.
>
> We propose both local and global pipelines to generate diversified instructions in InDET. These steps are illustrated in Sec. B and Sec. C. The objective and prompt in each pipeline can be summarized as follow:
>
> - **local prompt pipeline** (i.e., multi-modality pipeline)
> In this pipeline, we obtain diversified instructions of one single bbx in an image by querying LLAVA for multiple times. The prompts for LLAVA consist of visual and textual content. The visual content is the image with a red object bbx marked on it, and the textual content is empirically designed and provided in Table 4.
>
> - **global prompt pipeline** (i.e., single modality pipeline)
> In this pipeline, we obtain instructions related to multiple interested objects in a given image by LLAMA. The prompts for LLAMA consist of task description, in-context examples and current image information. The details of these elements are explained in **Response for Weakness 3**.
>
> To design appropriate prompts for LLAMA and LLAVA, we start from prompts commonly used in previous works or in the LLM community. We have tried some methods such as editing language expression, adding in-context examples, etc, to optimize the current prompts until achieving expected output.
>
> For services and cost of the procedure, thanks to the contributors of LLAMA and LLAVA, we can run those methods locally on our machines based on available implementations. And we add the details of hardware resource and time consumption cost of running the two methods in Appendix K.
>
> &nbsp;
>
> **Response for Question 2.**
>
> Yes. We will release our InDET dataset to benefit future ROD research.

---

### Official Review · Reviewer_Vtb3 · 2023-11-01

**Soundness:** 3 good
**Presentation:** 4 excellent
**Contribution:** 4 excellent
**Rating:** 6
**Confidence:** 4

**Summary:**

The authors propose InstructDET for referring object detection (ROD) that localizes objects based on user instructions. They leverage diverse instructions generated by vision-language models to create the InDET dataset. Based on InDET, their InstructDET outperforms existing methods on referring expression comprehension datasets and their own InDET test set.

**Strengths:**

- As the authors claim, InDET is the largest real-world REC dataset. It is expected that InDET will push the study of REC and its application.
- The dataset covers many real-world scenarios, including multiple object instruction.
- The proposed DROD model is simple yet effective. And it helps prove the quality of InDET.

**Weaknesses:**

- More experiments are expected. For instance, previous REC models should be trained on InDET and evaluated on RefCOCO/+/g.

**Questions:**

See Weakness.

---

> ### Author Response · Authors · 2023-11-22
> **Add experiments on previous REC models as suggested**
>
> Dear Reviewer Vtb3,
> We thank for your comments and answer the raised questions below.
>
> **Weakness 1. More experiments are expected. For instance, previous REC models should be trained on InDET and evaluated on RefCOCO/+/g.**
>
> We have conducted evaluations on previous REC models under the suggested configurations. The results are shown on the below table.
> | Method | backbone | Data | Refcoco val | Refcoco testA | Refcoco testB| Refcoco+ Val | Refcoco+ testA | Refcoco+ testB | Refcocog val | Refcocog test |
> | :-----| :----: | :----: | :----: | :----: | :----: | :----: | :----: | :----: | :----: | :----: |
> | MDETR | Res101 | RefC | 82.98 | 83.84 | 77.83 | 77.94 | 80.28 | 68.33 | 78.30 | 77.37 |
> | MDETR | Res101 | InDET | 84.67 | 85.34 | 80.51 | 79.17 | 83.39| 71.49 | 78.95 | 77.84 |
> | G-DINO | SwinB | RefC | 88.76 | 91.07 | 85.23 | 79.41 | 86.31 | 70.64 | 82.79 | 81.35 |
> | G-DINO | SwinB | InDET | 89.04 | 91.22 | 87.42 | 79.97 | 86.72 | 73.85 | 86.26 | 83.12 |
>
> For each REC method, we perform two experiments. One is to finetune the model on RefCOCO/+/g datasets directly. The other one is to pretrain on InDET and then finetune on RefCOCO/+/g datasets. For a fair comparison, here our InDET dataset only utilizes RefCOCO/+/g datasets but with diversified instructions.
>
> For MDETR, the original MDETR performs pre-processing on the expression to extract root of the expression using a dependency parser, and train the model to predict bbox match the root token only. This training framework is unsuitable for ROD task where the explicit name of the referring target might not show up in the expression. Inspired by UNINEXT, we replace the predefined linear class embedding layer with a visual-language alignment module to compute the matching extent between boxes and the whole expression. For G-DINO, the results are produced by the model retraining. The above table shows that the performance of existing REC models (i.e., MDETR and G-DINO) is improved by training with our InDET dataset, compared to the results by training with RefCOCO/+/g datasets.

---

### Author Response · Authors · 2023-11-23
**Response to AC and All Reviewers**

Dear AC and all reviewers,

We sincerely appreciate your time and efforts in reviewing our paper. We are glad to find that reviewers recognized the following merits of our work:

- **Innovative and meaningful instruction generation methods**：It would be of great interests to the community in general. The method taken from this paper seems to quite novel. The paper proposes a novel and interesting way to produce rich visual descriptions of objects and their relationships using LLM and VLM. [uAXw, nJxy]
- **The driving significance of the InDET dataset for Visual Grounding tasks**:  InDET is the largest real-world REC dataset. It is expected that InDET will push the study of REC and its application. This approach is scaleable which means that the results can continue to improve with more bbox data. The proposed dataset may be a good resource for future research in visual grounding tasks. [Vtb3,  nJxy, n3DU]
- **InDET dataset's remarkable impact on instruction generalization ability in models**:  The proposed DROD model is simple yet effective. And it helps prove the quality of InDET. The resultant model trained on the curated dataset seems to outperform prior arts by a large margin, even with the same network architecture. [Vtb3, uAXw]
- The popularity of our Clip-based instruction filtering methods: The "vision and language mapping" via CLIP as a verification step is particularly interesting. I like the approach for removing hallucination from the VLM model using vision language matching for CLIP. [uAXw, nJxy]

We also thank all reviewers for their insightful and constructive suggestions, which help further improve our paper. In addition to the pointwise responses below, we summarize the major revision in the rebuttal according to the reviewers’ suggestions.

- **Additional experiments**: ([Vtb3, nJxy, n3DU]) We have added experiments on training existing methods (i.e. MDETR, G-DINO) on our InDET dataset and testing on the RefCOCO/+/g datasets. The InDET dataset is generated based on exactly the same images and targets of RefCOCO. The performance of existing models on the RefCOCO/+/g dataset is improved by training with our InDET dataset. These results indicate that our generated instructions improve ROD of existing models. Moreover, we conduct a detailed evaluation in Sec. 6 to show that our instructions can improve the logic reasoning of existing models across image and text domains.
- **Manuscript update**: ([Vtb3, uAXw, nJxy, n3DU]) We modify presentations of the manuscript including grammar error correction and clearifying notifications. Also, we have added the footnotes to elucidate our usage of foundation models. In Appendix B and C, we provide the details of the two generation pipelines (i.e., local and global prompt pipelines). In Appendix I and J, we add post-processing, additional method comparisons and experiment results. The time consumption cost of the pipelines are added in Appendix K. Our revision is marked in blue.

We hope our pointwise responses below could address the raised concerns. We thank for all reviewers’ efforts and time again.

&nbsp;

Best,

Authors

---

### Meta-Review · Area_Chair_BLej · 2023-12-17

**Metareview:**

This  InstructDET method is designed for referring object detection (ROD) that localizes objects based on user instructions and provides a new pipeline based on visual foundation models, which can cover quite a lot of applications. Before rebuttal, reviewers raised a lot issues over missing experiments. During the discussion stage, authors add adequate experiments and explanations. Most of reviewers give positive comments. Reviewer n3DU is concerned about the quality of the dataset, which authors response well. The ACs thus decided to accept it.

**Justification For Why Not Higher Score:**

The dataset can be better checked to improve the quality.

**Justification For Why Not Lower Score:**

This proposed dataset and pipeline of building referring object detection via user instructions is quite inspiring to the community.

---

> ### Public Comment · ~Christopher_Kanan1 · 2024-06-13
> **Omitted Related Work: Visual Query Detection**
>
> Back in 2019, we published VQDv1 in NAACL, which has a lot of similar properties to the dataset in this paper (we called it visual query detection), where the system must output 0-K bounding boxes per prompt.
>
> Here is the website: https://www.manojacharya.com/vqd/
>
> NAACL paper: https://arxiv.org/abs/1904.02794

---

> > ### Public Comment · ~Yibing_Song1 · 2024-06-14
> >
> > The language-based detection methods have arisen before and we may not fully cover them in our paper. Thanks for the reminder and we will incorporate VQD in our next version.

---

### Decision · Program_Chairs · 2024-01-16

Accept (poster)